# Efficient Sharpness-Aware Minimization for Molecular Graph Transformer Models

**Yili Wang**[1], **Kaixiong Zhou**[2], **Ninghao Liu**[3], **Ying Wang**[4], **Xin Wang**[1] *

[1]School of Artificial Intelligence, Jilin University, China
[2]Institute for Medical Engineering & Science, Massachusetts Institute of Technology, USA
[3]School of Computing, University of Georgia, USA
[4]College of Computer Science and Technology, Jilin University, China
`wangyl21@mails.jlu.edu.cn, kz34@mit.edu,`
`ninghao.liu@uga.edu, {wangying2010, xinwang}@jlu.edu.cn`

## Abstract

Sharpness-aware minimization (SAM) has received increasing attention in computer vision since it can effectively eliminate the sharp local minima from the training trajectory and mitigate generalization degradation. However, SAM requires two sequential gradient computations during the optimization of each step: one to obtain the perturbation gradient and the other to obtain the updating gradient. Compared with the base optimizer (e.g., Adam), SAM doubles the time overhead due to the additional perturbation gradient. By dissecting the theory of SAM and observing the training gradient of the molecular graph transformer, we propose a new algorithm named GraphSAM, which reduces the training cost of SAM and improves the generalization performance of graph transformer models. There are two key factors that contribute to this result: (i) *gradient approximation*: we use the updating gradient of the previous step to approximate the perturbation gradient at the intermediate steps smoothly (**increases efficiency**); (ii) *loss landscape approximation*: we theoretically prove that the loss landscape of GraphSAM is limited to a small range centered on the expected loss of SAM (**guarantees generalization performance**). The extensive experiments on six datasets with different tasks demonstrate the superiority of GraphSAM, especially in optimizing the model update process. The code is in: https://github.com/YL-wang/GraphSAM/tree/graphsam.

## 1 Introduction

Biochemical molecular property prediction is one of the essential tasks for many applications, including drug discovery (Withnall et al., 2020; Wu et al., 2018; Ye et al., 2022) and molecular fingerprint design (Honda et al., 2019; Kearnes et al., 2016). Motivated by the recent progress in natural language processing and computer vision, graph transformer models have shown promising results for molecular property prediction by treating molecular structures as graphs (Rong et al., 2020; Chen et al., 2021a). The transformers learn the global interaction of every node to capture the underlying structure, improving the molecular property classification performance.

Without the hand-crafted features or inductive biases encoded in the neural architecture, it is widely observed that the transformer is prone to converge to sharp local minima, where the loss value changes quickly in the neighborhood around model weights (Du et al., 2021; Zhou et al., 2022b; Chen et al., 2021b; Andriushchenko & Flammarion, 2022; Zhou et al., 2021b). The sharp local minima are highly correlated with significant generalization errors in various domains. To mitigate the sharpness, existing studies on transformers for molecular property prediction use large-scale pre-training, while the pre-training requires extensive resources and expert knowledge in constructing the informative self-supervised learning tasks (Rong et al., 2020; Chithrananda et al., 2020; Ying et al., 2021; Liu et al., 2021b). These laborious requirements prevent engineers from directly applying the transformer for any downstream application.

Recently, sharpness aware minimization (SAM) (Foret et al., 2020) has been proposed to explicitly smooth the sharp local minima during model training like pre-training. Nevertheless, SAM requires two forward and backward propagations at each step: one to obtain the worst-case adversarial gradient

---

*Correspondence to: Xin Wang

for perturbing weights named **perturbation gradient** and the other to obtain the **updating gradient** for the training model. Compared with the base optimizer, **SAM doubles the time overhead due to the extra computation of the perturbation gradient.** In order to improve the efficiency of model training, some efficient SAM variants (AE-SAM, RST, ESAM, etc.) (Jiang et al., 2023; Zhao et al., 2022a; Du et al., 2021) have been invented. They seek to enhance efficiency through "generic" techniques (e.g., Stochastic judgment gradient calculation) that can be employed with "any" model and have been successful in Computer Vision. Whether they still maintain a high standard when they step out of their comfort area, on the contrary, the answer is no. As shown in Fig. 1, compared to the base optimizer of Adam (the second bar), the performance of the pre-training-free Grover model with SAM has a significant improvement (the third bar). However, other efficient SAM variants have minimal or even counterproductive performance. The reason is that *they overlooked or omitted the principles of SAM's success: gradient direction and gradient size.* This is the core aspect that motivates the current work.

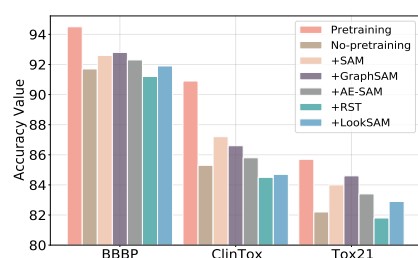

Figure 1: The results of GROVER with different strategies on three datasets.

In this paper, we aim to improve the **efficiency** of SAM while maintaining **generalization performance** of graph transformer models on molecular property prediction tasks. One of the existing straightforward solutions to speed up SAM is to compute the perturbation gradient periodically, e.g., RST (Zhao et al., 2022a), which unfortunately leads to poor empirical results. In the experiments, we observe the perturbation gradient direction is similar to the indispensable updating gradient direction from the last step. Based on this discovery, we develop GraphSAM to reuse the updating gradient of the previous step and approximate the perturbation gradient with free lunch. Specifically, GraphSAM contains two sequential gradient computations at each step: one to obtain the updating gradient by one forward and backward propagation and the other to use it to prepare the perturbation gradient for the next step. The perturbation gradient is periodically re-anchored and gradually smoothed with the updating gradient at the intermediate steps. By relieving the perturbation gradient computation overhead, GraphSAM has comparable efficiency with the traditional optimizers. More importantly, we verify through formula derivation and experiments that GraphSAM has the ability to compete with SAM. GraphSAM can converge to the flat minima, as shown in Fig. 2. Our contribution is in the following four aspects:

- We observe the sharp local minima in a convergence of the graph transformer model for the molecular property prediction problem. Similar to other domains, the sharp local minima result in poor generalization performance.

- We propose GraphSAM to relieve the computation overhead of SAM. In particular, instead of computing the perturbation gradient at every step, GraphSAM computes it periodically and uses the updating gradient of the previous step to approximate it smoothly.

- We experimentally observe the approximated perturbation gradient is close to the ground-truth, and theoretically prove the loss landscape is constrained within a small bound centered at the desired loss of SAM.

- The extensive experiments on the benchmark molecular datasets show that GraphSAM achieves comparable or even outperforming results, and frees the expensive pre-training. The time overhead is marginal compared with SAM.

## 2 RELATED WORK

**Sharp Local Minima.** Sharp local minima can primarily affect the generalization performance of deep neural networks (Izmailov et al., 2018; Jastrzebski et al., 2017; Guo et al., 2022; Zhou et al., 2022a; 2023; Wang et al., 2022b; 2020). Recently, many studies have attempted to explore how to solve the optimization problem by locating the parameters in flat minima instead of sharp local minima (Zhou et al., 2020; Jiang et al., 2019; Moosavi-Dezfooli et al., 2019; Zhou et al., 2021a; Liu et al., 2020; Wen et al., 2018). At the same time, it is shown through a large number of experiments that there is a strong correlation between sharpness and generalization error at various hyperparameter settings (Jiang et al., 2019). This motivates the idea of minimizing sharpness during training to

improve standard generalization, resulting in sharpness-aware minimization(SAM) (Foret et al., 2020).

**Sharpness-Aware Minimization.** SAM has been successfully used in areas such as image classification (Foret et al., 2020; Du et al., 2021; Liu et al., 2022; Zhao et al., 2022b) and natural language processing (Bahri et al., 2021). Especially, LookSAM improves computational efficiency by eliminating the calculation of the updating gradient (Liu et al., 2022). SAF (Du et al., 2022) suggests an innovative trajectory loss that mitigates abrupt decreases in the loss at sharp local minima during the weight update trajectory to reduce the time loss. Nevertheless, most of the work still ignores the fact of SAM's double overhead (Damian et al., 2021; Kwon et al., 2021; Wang et al., 2022a; Li & Giannakis, 2023) and no studies of SAM are available in the graph domain. This forces us to propose the GraphSAM algorithm in the field of molecular graphs, which retains the generalization ability of SAM while improving computational efficiency.

**Molecular Graph Representation Learning.** In recent years, Steven et al.Kearnes et al. (2016); Xiong et al. (2019) have tried applying GNNs to molecular characterization learning to better utilize the structural information of molecules. Nevertheless, they have poor generalization ability to new-synthesized molecules. Recently, a great deal of pre-training has been done on graphs to capture the rich information in molecular graphs (Hu et al., 2019; 2020; Rong et al., 2020; Honda et al., 2019). In contrast, the pre-training-free graph transformer models are prone to overfit and have poor generalization performance, thus some recent studies utilize contrastive learning and data augmentation methods to solve the problem (Xia et al., 2022; Zhang et al., 2020). However, these methods have a considerable training time and are complex to operate.

## 3    PROBLEM STATEMENT AND SAM

### 3.1    MOLECULAR PROPERTY PREDICTION

A molecular structure can be represented by an attributed graph $G = (\mathcal{V}, \mathcal{E})$, where $\mathcal{V}$ and $\mathcal{E}$ denote the atoms (nodes) and chemical bonds (edges) within the molecule, respectively. The initial feature of node $v$ is denoted as $x_v$, and the initial feature of edge $(u, v)$ is $e_{uv}$. The set of neighbors of node $v$ is denoted as $\mathcal{N}_v$. We consider two categories of molecular property prediction tasks: *graph classification* and *graph regression*. Both of them are given a set of molecular graphs $\mathbb{G} = \{G_1, \cdots, G_N\}$ and their labels/targets $\{y_1, \cdots, y_N\}$, where the molecular property prediction task is to infer the label/target of a new graph.

### 3.2    SHARP LOCAL MINIMA IN TRANSFORMER

**Weight Loss Landscape.** Existing studies have demonstrated that converging to a flat region in the loss landscape can improve the generalization performance of DNNs (Li et al., 2018; Du et al., 2021; Xue et al., 2021; Juan et al., 2023; Yang et al., 2022). To understand model convergence, we visualize the loss landscape of GROVER trained on the BBBP dataset in Fig. 2. The training loss (y-axis) is defined as $\mathcal{L}(\theta + \phi D_\theta)$ (e.g., cross-entropy loss), where $\theta$ denotes the best-trained model parameters at the convergence of GROVER. $D_\theta$ is the random gradient directions sampled from Gaussian distribution, and $\phi$ (x-axis) controls the scalar size of the step that moves by $D_\theta$ to get the perturbation parameters, which is called the best parameter's neighborhood. In Fig. 2, we can observe that the loss landscape of Adam is much sharper, i.e., its neighborhood parameters' training loss is much larger than SAM

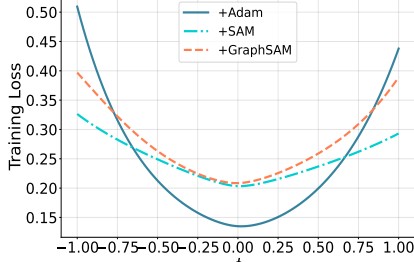

Figure 2:    The loss landscape of GROVER with different optimizers.

and GraphSAM. The over-parameterized pre-training-free graph transformer model is located in the sharp local minima.

**Generalization Performance.** Transformer models rely on massive pre-training (Rong et al., 2020; Yuan et al., 2021; Liu et al., 2021a; Touvron et al., 2021; Sun et al., 2022) to mitigate the sharp local minima problem, such as GROVER (Rong et al., 2020). They are demanding data and computation resources, and require laborious tuning and expert knowledge in designing the pre-training tasks. When the pre-training process is removed, the models often fall into the sharp local minima, which significantly affects the generalization performance of graph transformer models. As shown in Fig. 1,

we use the test accuracy value to express the models' generalization performance, and the GROVER performance significantly deteriorates once pre-training is ablated.

### 3.3 SHARPNESS-AWARE MINIMIZATION

In order to improve generalization and free model construction from large-scale pre-training, Foret et al. (Foret et al., 2020) propose the SAM algorithm to seek parameter values whose entire neighborhoods have both low loss and low curvature. Formally, SAM trains a transformer by solving the following min-max optimization problem:

$$\min_{\theta} \max_{\|\hat{\epsilon}\|_2 \leq \rho} \mathcal{L}_{\mathbb{G}}(\theta + \hat{\epsilon}), \tag{1}$$

where $\rho$ is the size of the gradient stepping ball, $\mathbb{G}$ is the training dataset, and $\theta$ denotes the model weight parameters. Here we omit the regularization term for simplicity. At training step $t$, SAM solves the min-max problem by the following iterative process:

$$\epsilon_t = \nabla_\theta \mathcal{L}_{\mathbb{G}}(\theta_t), \quad \hat{\epsilon}_t = \rho \cdot \frac{\epsilon_t}{\|\epsilon_t\|_2}, \quad \omega_t = \nabla_\theta \mathcal{L}_{\mathbb{G}}(\theta_t + \hat{\epsilon}_t), \quad \theta_{t+1} = \theta_t - \eta_t \cdot \omega_t. \tag{2}$$

For the inner optimization, SAM calculates the *perturbation gradient* $\epsilon_t$ with a complete forward and backward propagation and then normalizes $\epsilon_t$ within the $\rho$-ball to get $\hat{\epsilon}_t$. The normalized gradient $\hat{\epsilon}_t$ is applied to update the model as shown by the inner maximization problem in equation 1. For the outer optimization, SAM obtains the *updating gradient* $\omega_t$ with another complete forward and backward propagation. Then $\omega_t$ is used to update the model with the learning rate $\eta_t$ towards the expected smooth minima. SAM consumes double overhead due to the extra computation of perturbation gradient compared with the base optimizer.

## 4 GRAPHSAM

### 4.1 UNDERSTANDING OF THE GRADIENTS

To optimize the computational efficiency of SAM, we provide deeper insights into the perturbation gradient $\epsilon_t$ and the updating gradient $\omega_t$. Particularly, we introduce two key observations as motivation from experiments conducted on the BBBP dataset with two graph transformer models.

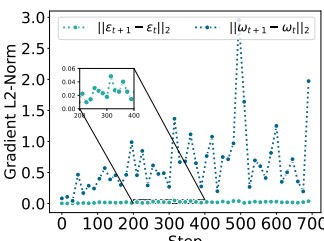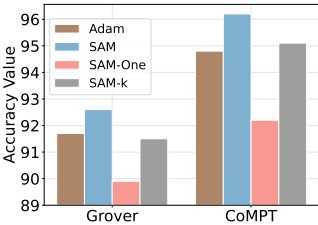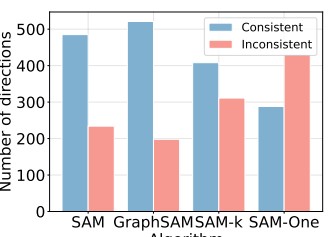

Figure 3: Illustration on the observation of gradient on GROVER and CoMPT with BBBP dataset. (a) Gradient variation during training. (b) Accuracy of Adam, SAM, SAM-One, and SAM-$k$. (c) The similarity direction numbers of $\epsilon_{t+1}^S$ and $\omega_t^S$ in SAM , or the similarity direction numbers of $\epsilon_t$ and $\epsilon_t^S$ in GraphSAM, SAM-K, and SAM-One.

**Observation 1** (*The change of $\epsilon_t$ and $\omega_t$ during the iterative training process*). As shown in Fig. 3(a), we use $\|\epsilon_{t+1} - \epsilon_t\|_2$ and $\|\omega_{t+1} - \omega_t\|_2$ to describe the changing degrees of perturbation gradient $\epsilon_t$ and the updating gradient $\omega_t$, respectively. It is observed that $\omega_t$ changes more drastically and has a broader range during the training process. In contrast, the variation of $\epsilon_t$ is relatively minor, which can lead us to focus on the perturbation gradient to improve the computational loss of the model. We also observe $\|\omega_t\|_2 \gg \|\epsilon_t\|_2$ in Fig. 9, which is useful for the subsequent proof of the theory.[1]

According to Observation 1 where the perturbation gradient changes gently, **an intuitive solution** to reducing the time complexity of SAM is to compute $\epsilon_t$ only once at the beginning, and then reuse

---

[1]The observation is unique to the domain of molecular graphs.

the initial $\epsilon_t$ at the following training steps. Mathematically, we name such an efficient solution as SAM-One, where the perturbation gradient is given by $\epsilon_t = \epsilon_0$, for $t \geq 1$. Notably, SAM-One can be regarded as a special case of SAM-$k$ (Liu et al., 2022), where the perturbation gradient is computed every $k$ step. Although the computational efficiency is improved, the performances of both SAM-One and SAM-$k$ deteriorate significantly as shown in Fig. 3(b). For example, SAM-One consistently delivers the worst generalization performance. Compared with SAM on the CoMPT model, the molecule property classification accuracy of SAM-One drops from 96.2% to 92.2%. One possible reason is that the influence of $\epsilon_t$ fluctuation is non-negligible, as shown in the small window of Fig. 3(a). The old perturbation gradient obtained many steps before, fails to accurately reflect the scale of current $\epsilon_t$. This experimental observation demonstrates that the vanilla algorithm of SAM-One or SAM-$k$ does not adapt well to the molecular property prediction problem. Thus, we further study the correlation between the perturbation gradient and the updating gradient as follows.

**Definition 1** (Similarity metric). *We use the cosine function to measure the similarity between two vectors, denoted as* $\cos(\cdot, \cdot)$. *The vector intersection angle is denoted as* $\angle$, *e.g., if* $\cos(a, b) = 1$, *then* $\angle_{ab} = 0°$.

**Definition 2** (Gradient direction similarity). *Given the perturbation gradient* $\epsilon_t$ *and the updating gradient* $\omega_t$, *if* $0 \leq \cos(\epsilon_t, \omega_t) \leq 1$, *we call* $\epsilon_t$ *and* $\omega_t$ *are **consistent**; and if* $-1 \leq \cos(\epsilon_t, \omega_t) < 0$, *then* $\epsilon_t$ *and* $\omega_t$ *are **inconsistent**. That is, the angle* $\angle$ *between consistent vectors satisfies* $0° \leq \angle \leq 90°$.

**Observation 2** *(Similarity between* $\epsilon_{t+1}$ *and* $\omega_t$*).* As shown in Fig. 3(c), we divide the similarity scores between the $(\epsilon_{t+1}, \omega_t)$ pair of SAM into two categories: ***consistent*** pairs with $0 \leq \mathrm{Cos}(\epsilon_{t+1}, \omega_t) \leq 1$, and ***inconsistent*** pairs with $-1 \leq \cos(\epsilon_{t+1}, \omega_t) < 0$. We observe that the gradient direction of $\omega_t$ is prone to be close to that of $\epsilon_{t+1}$ during the training process. *In particular, consistent pairs account for 67.45% in the overall training process.*

The above observations motivate us to approximate the perturbation gradient with the updating gradient from the last step, in order to save time for the model optimization computation.

## 4.2 AN EFFICIENT SOLUTION

In the following, we propose a novel optimization algorithm, GraphSAM, to address the above challenges. The innovation of GraphSAM has the following two aspects: perturbation gradient approximation and gradient ball's size ($\rho$) scheduler.

**Perturbation gradient approximation.** GraphSAM contains two sequential gradient computations at each step: one to obtain the updating gradient by forward and backward propagation, and the other to use the updating gradient to prepare the perturbation gradient for the next step. Indeed, at training step $t$, we have:

$$\epsilon_0 = \nabla_\theta \mathcal{L}_\mathbb{G}(\theta_0), \quad \hat{\epsilon}_t = \rho \cdot \frac{\epsilon_t}{||\epsilon_t||_2}, \quad \omega_t = \nabla_\theta \mathcal{L}_\mathbb{G}(\theta_t + \hat{\epsilon}_t),$$
$$\theta_{t+1} = \theta_t - \eta_t \cdot \omega_t, \quad \epsilon_{t+1} = \beta \cdot \epsilon_t + (1 - \beta) \cdot \omega_t / \parallel \omega_t \parallel_2 . \tag{3}$$

Like the SAM algorithm, at training step $t = 0$, we need an additional forward and backward propagation to compute the perturbation gradient $\epsilon_0$. Then we project the perturbation gradient within the $\rho$-ball to get $\hat{\epsilon}_t$ and the neighboring parameters $(\theta_t + \hat{\epsilon}_t)$ of $\theta_t$. With another forward and backward propagation from $(\theta_t + \hat{\epsilon}_t)$, we can get the updating gradient $\omega_t$ to update the model. Unlike the SAM algorithm, at training step $t + 1$, GraphSAM uses the idea of moving average to prepare $\epsilon_{t+1}$ by combining the $\epsilon_t$ with $\omega_t$ of the previous step, and the hyperparameter $\beta \in [0, 1)$ controls the exponential decay rate of the moving average. The moving average method can effectively retain the information of the previous steps of $\omega_t$, and the initial information of $\epsilon_0$. The $\epsilon_{t+1}$ acquired by this method can be approximated with the ground truth. Meanwhile, it prevents computing the perturbation gradient at each step with additional forward and backward propagation, which increases the efficiency of SAM for model optimization.

Compared with SAM in equation 2, instead of exactly calculating perturbation gradient with extra forwarding and backward, GraphSAM approximates $\epsilon_{t+1}$ for the next step based on the above two observations: $\epsilon_t$ changes slowly and $\epsilon_{t+1}$ positively relates to $\omega_t$. As the training step increases, $\epsilon_{t+1}$ obtained by moving average may gradually deviate from the ground-truth exactly computed by

SAM. Therefore, to reduce the error, the perturbation gradient $\epsilon_{t+1}$ is periodically re-anchored, e.g., the perturbation gradient is recalculated at the first-step of each epoch. This ultimately makes the direction of $\epsilon_{t+1}$ obtained by the moving average is keeping close to the ground-truth.

To evaluate the approximation performance, we compare the approximated perturbation gradients obtained by SAM-One, SAM-$k$, and GraphSAM in Fig. 3(c). Specifically, we measure their cosine similarity scores with the ground-truth perturbation gradient obtained by SAM. As shown in Fig. 3(c), $consistent pairs$ accounts for 72.46% of the total training process in GraphSAM. The $consistent pairs$ statistics of SAM-One (40.05%) and SAM-$k$ (56.74%) algorithms are much worse than GraphSAM. The results show that our GraphSAM provides a better solution to approximate the perturbation gradient.

**Gradient ball's size** ($\rho$) **scheduler.** As seen from equation 3, the projected perturbation gradient size $\hat{\epsilon}_t$ is determined not only by the perturbation gradient $\epsilon_t$, but also by the size of gradient ball $\rho$. The models' generalization ability is correlated intimately with $\rho$ (Kwon et al., 2021), as shown in Table 8 in the experiment. In general, the size of the perturbation gradient should change with the training process. The early stages of training require a larger perturbation gradient size to enable graph transformer models to handle the much sharper minima cases. When the model converges at the later stages of training, the perturbation gradient should decrease or even converge to 0, allowing the model to fall into a flatter region. Therefore, motivated by the learning rate scheduling (e.g., StepLR (Brownlee, 2016)), we propose a simple but effective gradient ball's size scheduler as below:

$$\rho_{\text{new}} = \rho_{\text{initial}} * \gamma^{\text{epoch}/\lambda}, \tag{4}$$

where the hyperparameter $\lambda$ controls the update rate of $\rho$, i.e., how many epochs for one update. The hyperparameter $\gamma$ controls the modification scale in $\rho$. Summarizing the above, the GraphSAM algorithm is shown in Appendix.

### 4.3 GRADIENT APPROXIMATION ANALYSIS

In this section, we verify that the perturbation gradient of GraphSAM approximates the ground truth. Meanwhile, we prove the loss landscape of GraphSAM is limited to a small range centered on the expected loss of SAM.

Before going to the mathematical proof, we first illustrate GraphSAM and SAM in Fig. 4. Considering the model parameter learning from $\theta_t$ to $\theta_{t+1}$, GraphSAM first approximates the perturbation gradient and then adversarially perturbs the graph transformer model to the neighborhood, where the model is finally updated to $\theta_{t+1}$ with a complete forward and backward propagation. In particular, the perturbation gradient $\epsilon_t^{\text{G}}$ of GraphSAM is obtained from the moving average between $\epsilon_{t-1}$ and $\omega_{t-1}$ as shown by the last equation in equation 3. The projected perturbation gradient $\hat{\epsilon}_t^{\text{G}}$ practically used to perturb model to neighborhood $\theta_t^{\text{adv}}$, is obtained by mapping $\epsilon_t^{\text{G}}$ over the gradient ball. Based upon $\theta_t^{\text{adv}}$, we compute the updating gradient $\omega_t$ by a forward and backward propagation to update $\theta_t \rightarrow \theta_{t+1}$. Therefore, each training step of GraphSAM only requires the same tensor flowing as the base optimizer. In contrast, as illustrated in Fig. 4, SAM takes another complete propagation to obtain $\epsilon_t^{\text{S}}$ and its projected one $\hat{\epsilon}_t^{\text{S}}$. Since $\hat{\epsilon}_t^{\text{S}}$ is exactly estimated, we treat it

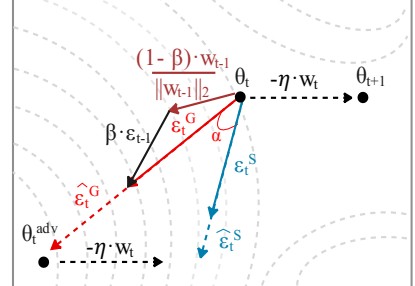

Figure 4: Visualization of GraphSAM. The black dashed line is obtained from GraphSAM's updating gradient which targets the model's parameters $\theta$ moving to a flatter region.

as the ground truth compared with the approximated perturbation gradient $\hat{\epsilon}_t^{\text{G}}$ used in GraphSAM.

**Conjecture 1.** *Let $\hat{\epsilon}_{\text{S}}$ and $\hat{\epsilon}_{\text{G}}$ denote the perturbation weights of SAM and GraphSAM, respectively, where we ignore the subscript of t for the simple representation. Suppose that $\frac{\omega}{||\omega||_2} \gg \epsilon$ as empirically discussed in Observation 1[2], and $||\hat{\epsilon}_{\text{S}}||_2 < ||\hat{\epsilon}_{\text{G}}||_2$ for $\rho > 0$ designating $\hat{\epsilon}_{\text{S}}$ as the ground-truth. We have:*

$$\max_{||\hat{\epsilon}_{\text{S}}||_2 \leq \rho} \mathcal{L}_{\mathbb{G}}(\theta + \hat{\epsilon}_{\text{S}}) \leq \max_{||\hat{\epsilon}_{\text{G}}||_2 \leq \rho} \mathcal{L}_{\mathbb{G}}(\theta + \hat{\epsilon}_{\text{G}}).$$

---

[2]This assumption is obtained from Fig. 9

We list the detailed proof in Appendix A.2. From the generalization capacity perspective, it is observed that the adversarially perturbed loss of SAM is upper bounded by that of GraphSAM. Recalling the min-max optimization problem of SAM in equation 1, if we replace the inner maximum objective from $\mathcal{L}_{\mathbb{G}}(\theta + \hat{\epsilon}_{\mathrm{S}})$ to $\mathcal{L}_{\mathbb{G}}(\theta + \hat{\epsilon}_{\mathrm{G}})$, the graph transformer is motivated to smooth a worse neighborhood loss in the loss landscape. In other words, the proposed GraphSAM aims to minimize a rougher neighborhood loss, whose value is theoretically larger than that of SAM, and obtain a smoother landscape associated with the desired generalization.

**Conjecture 2.** *Let* $||\mathcal{L}_{\mathbb{G}}(\theta + \hat{\epsilon}_{\mathrm{G}}) - \mathcal{L}_{\mathbb{G}}(\theta + \hat{\epsilon}_{\mathrm{S}})||$ *denote the gap of generalization performance between SAM and GraphSAM. It is positively correlated with the gradient approximation error as below:*

$$\| \mathcal{L}_{\mathbb{G}}(\theta + \hat{\epsilon}_{\mathrm{G}}) - \mathcal{L}_{\mathbb{G}}(\theta + \hat{\epsilon}_{\mathrm{S}}) \| \propto \| \hat{\epsilon}_{\mathrm{G}} - \hat{\epsilon}_{\mathrm{S}} \| .$$

We list the detailed proof in Appendix A.1. Given the above-constrained loss bias and the limited training epochs, the optimization algorithm of GraphSAM is prone to converge to the area neighboring that of SAM, where the loss landscape is smooth and desired for good generalization. Indeed, we can compute the loss bias by:

$$\| \hat{\epsilon}_{\mathrm{G}} - \hat{\epsilon}_{\mathrm{S}} \| \leq \| \alpha \cdot \hat{\epsilon}_{\mathrm{G}} \| = \alpha \cdot \| \rho \cdot \frac{\epsilon_{\mathrm{G}}}{\| \epsilon_{\mathrm{G}} \|} \|, \tag{5}$$

where $\| \alpha \cdot \hat{\epsilon}_{\mathrm{G}} \|$ denotes the arc length between gradients $\hat{\epsilon}_{\mathrm{G}}$ and $\hat{\epsilon}_{\mathrm{S}}$. When the intersection angle $\alpha$ (or the size of gradient ball $\rho$) is close to 0, $\hat{\epsilon}_t^{\mathrm{G}}$ is infinitely approximated to $\hat{\epsilon}_t^{\mathrm{S}}$. Conjecture 2 with equation 5 shows GraphSAM's loss landscape is constrained within a small bound centered at the desired loss of SAM.

As the training step $t$ increases, the intersection angle $\alpha$ between $\hat{\epsilon}_t^{\mathrm{G}}$ and $\hat{\epsilon}_t^{\mathrm{S}}$ gets wider, and the generalization performance of GraphSAM may reduce significantly. As seen from equation 5, in addition $\alpha$, the size of gradient ball $\rho$ plays another key role in determining the gradient approximation error. Therefore, to ensure GraphSAM has a similar generalization performance as SAM, we periodically reduce the size of $\rho$ by using equation 4. In summary, we use the moving average method to approximate the ground-truth, and control the size of $\rho$ to reduce the gap between $\hat{\epsilon}_t^{\mathrm{G}}$ and $\hat{\epsilon}_t^{\mathrm{S}}$. Both of them are the keys to keeping GraphSAM with good generalization performance.

## 5 EXPERIMENTS RESULTS

In this section, we evaluate the effectiveness of GraphSAM on two graph transformer models. Overall, we aim to answer three research questions as follows. **Q1:** Can GraphSAM improve the generalization performance of the graph transformer models? **Q2:** How effective is GraphSAM compared to SAM and other variants of SAM? **Q3:** How do each module and key hyperparameters of GraphSAM affect its efficiency and performance?

### 5.1 EXPERIMENT SETUP

**Datasets.** Following the settings of previous molecular graph tasks, we consider six public benchmark datasets: BBBP, Tox21, Sider, and ClinTox for the classification task, and ESOL and Lipophilicity for the regression task. We evaluate all models on a random split as suggested by MoleculeNet (Wu et al., 2018), and split the datasets into training, validation, and testing with a 0.8/0.1/0.1 ratio. More detailed statistics are provided in Appendix A.3. In addition, the **Backbone frameworks and baselines** and **Implementations** are described in detail in Appendix A.4.

### 5.2 PERFORMANCE ANALYSIS

#### 5.2.1 GENERALIZATION PERFORMANCE IMPROVEMENT

To answer the research question **Q1**, we apply GraphSAM to GROVER and CoMPT in a comprehensive comparison with the baseline approaches on different datasets. We make two key observations.

▷ Lacking the large-scale pre-training process, an over-parameterized graph transformer model does not necessarily work better than a GNNs-based model. As shown in Table 1, compared with GNNs-based models, GROVER (no pre-training) and CoMPT have even worse performance on most datasets than the GNNs-based CMPNN. **It fully illustrates that the over-parameterized transformer model tends to fall into the sharp local minima during the training process like Fig. 2.** The sharp local minima significantly affect the generalization performance of the graph transformer model.

Table 1: Prediction results of GraphSAM and baselines on six datasets. We used 5-fold cross-validation with random split and replicated experiments on each task five times. The mean and standard deviation of AUC or RMSE values are reported. $\boldsymbol{Bold}$ is the best optimizer, and $\underline{underlined}$ is the best model.

| Task | Graph Classification (ROC-AUC↑) | | | | Graph Regression (RMSE↓) | |
|---|---|---|---|---|---|---|
| Dataset | BBBP | Tox21 | Sider | ClinTox | ESOL | Lipophilicity |
| GCN | $0.690_{\pm 0.041}$ | $0.819_{\pm 0.031}$ | $0.623_{\pm 0.022}$ | $0.807_{\pm 0.044}$ | $0.970_{\pm 0.071}$ | $1.313_{\pm 0.149}$ |
| MPNN | $0.901_{\pm 0.032}$ | $0.834_{\pm 0.014}$ | $0.634_{\pm 0.014}$ | $0.881_{\pm 0.037}$ | $0.702_{\pm 0.042}$ | $1.242_{\pm 0.249}$ |
| DMPNN | $0.912_{\pm 0.037}$ | $0.845_{\pm 0.012}$ | $0.646_{\pm 0.020}$ | $0.897_{\pm 0.042}$ | $0.665_{\pm 0.060}$ | $1.159_{\pm 0.207}$ |
| CMPNN | $0.925_{\pm 0.017}$ | $0.837_{\pm 0.016}$ | $0.640_{\pm 0.018}$ | $0.918_{\pm 0.016}$ | $0.582_{\pm 0.055}$ | $0.633_{\pm 0.029}$ |
| GROVER | $0.917_{\pm 0.028}$ | $0.822_{\pm 0.019}$ | $0.649_{\pm 0.035}$ | $0.853_{\pm 0.043}$ | $0.639_{\pm 0.087}$ | $0.671_{\pm 0.047}$ |
| + SAM | $0.926_{\pm 0.022}$ | $0.840_{\pm 0.035}$ | $0.660_{\pm 0.043}$ | $\mathbf{0.872_{\pm 0.044}}$ | $\mathbf{0.619_{\pm 0.089}}$ | $0.662_{\pm 0.052}$ |
| + GraphSAM | $\mathbf{0.928_{\pm 0.016}}$ | $\mathbf{0.846_{\pm 0.012}}$ | $\mathbf{0.665_{\pm 0.038}}$ | $0.866_{\pm 0.051}$ | $0.625_{\pm 0.083}$ | $\mathbf{0.654_{\pm 0.056}}$ |
| CoMPT | $0.948_{\pm 0.025}$ | $0.828_{\pm 0.008}$ | $0.621_{\pm 0.013}$ | $0.914_{\pm 0.034}$ | $0.562_{\pm 0.071}$ | $0.618_{\pm 0.012}$ |
| + SAM | $\underline{\mathbf{0.962_{\pm 0.033}}}$ | $0.839_{\pm 0.006}$ | $0.643_{\pm 0.009}$ | $0.927_{\pm 0.025}$ | $0.517_{\pm 0.025}$ | $0.611_{\pm 0.015}$ |
| + GraphSAM | $0.961_{\pm 0.012}$ | $\mathbf{0.841_{\pm 0.004}}$ | $0.645_{\pm 0.013}$ | $\underline{\mathbf{0.937_{\pm 0.008}}}$ | $\underline{\mathbf{0.511_{\pm 0.018}}}$ | $\underline{\mathbf{0.608_{\pm 0.007}}}$ |

▷ GraphSAM and SAM can improve the generalization performance of transformer models that fall into sharp local minima. Compared with the vanilla model, adapting GraphSAM to each transformer model, it delivers the average improvements of 1.52% (GROVER), and 2.16% (CoMPT), respectively. SAM is 1.55% (GROVER) and 1.97% (CoMPT). We show the loss curves of GROVER and CoMPT with different optimizers in Fig. 5. SAM has the best generalization loss (the purple line), Graph-SAM is the second best (the green line), and Adam is the worst (the orange line). In particular, there is a large gap between Adam's training loss and testing loss. This is because

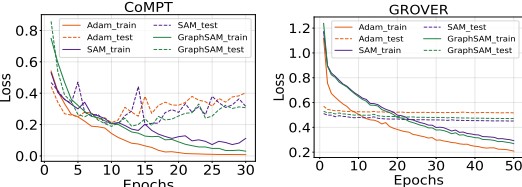

Figure 5: Loss curves of different optimizers on the BBBP dataset.

the over-parameterized pre-training-free graph transformer model is always prone to overfitting.

In addition, as shown in Fig. 2, we display the loss landscape and demonstrate that both GraphSAM and SAM have significantly improved the sharpness compared to Adam. **GraphSAM and SAM optimize the model update using the perturbation gradient to seek parameter values with low loss and curvature throughout the neighborhood.** It also prevents the parameters from falling into sharp local minima. Both GraphSAM and SAM can improve the model's generalization.

### 5.2.2 THROUGHPUT AND ACCURACY OF THE OPTIMIZERS

To answer the research question **Q2**, we summarize the model efficiency and generalization performance of different optimization algorithms in Table 7. In summary, we observe that:

▷ GraphSAM has a similar generalization performance to SAM while having high throughput. We compare the throughput and generalization performance of GraphSAM with other optimization algorithms in Table 2 and Table 7. Where throughput denotes the computational overhead which is quantified by graphs processed per second (Graphs/s), and the generalization performance is measured by test ROC-AUC/RMSE. The experimental results show that GraphSAM can improve the training speed up to 155.4% compared to SAM. The reason for this is that **we perform only one forward and backward propagation in most of the training steps, which will greatly improve the model's efficiency.** In terms of performance, GraphSAM is close to SAM and even better than SAM on some datasets in Table 1[3].

The other optimization algorithms also can improve efficiency but lose some generalization performance. For instance, although SAM-$k$ and SAM-One have a high throughput, their generalization performance is poor. This is because though they also periodically compute the perturbation gradient, it differs significantly from the ground-truth gradient at other training times, eventually affecting the model's generalization performance. LookSAM is an optimization algorithm designed for the

---

[3]Under the optimal parameters, SAM theoretically has the best performance

transformer model in the computer vision domain. Unlike our algorithm, it maintains the perturbation gradient by the updating gradient. Due to the design of LookSAM, it does not conform to the properties of the molecular graph model. Therefore, it is inferior to train the model based on the original hyperparameter. LookSAM has been tuned by us ($\rho = 0.0001, \alpha = 0.2, k = 8$), but its performances are only the same as Adam's in all datasets, and LookSAM takes more time. In addition, AE-SAM and RST compute the update gradient by different strategies, respectively. Because of the determination problem of the strategies, their computational consumption is very dissimilar in different models and data. There is also no guarantee that the generalization performance will not be affected. Please see Appendix A.4 and A.5 for details.

Table 2: Classification accuracy and training speed.

| | BBBP | |
|---|---|---|
| **GROVER** | ROC-AUC↑ | Throughput |
| Adam | 0.917 | 362 |
| SAM | 0.926 | 201(100.0%) |
| SAM-One | 0.899 | 330(164.2%) |
| SAM-$k$ | 0.915 | 287(142.7%) |
| LookSAM | 0.919 | 266(132.3%) |
| AE-SAM | 0.923 | 291(144.8%) |
| RST | 0.912 | 312(155.2%) |
| GraphSAM | **0.928** | 272(135.3%) |
| **CoMPT** | ROC-AUC↑ | Throughput |
| Adam | 0.948 | 218 |
| SAM | **0.962** | 112(100.0%) |
| SAM-One | 0.922 | 199(177.6%) |
| SAM-$k$ | 0.941 | 183(163.4%) |
| LookSAM | 0.952 | 161(143.8%) |
| AE-SAM | 0.955 | 178(158.9%) |
| RST | 0.940 | 186(166.1%) |
| GraphSAM | 0.961 | 174(155.4%) |

### 5.2.3 ROLES OF GRAPHSAM MODULES.

To answer the research question **Q3**, we conduct experiments on multiple graph transformer models and datasets for Adam, SAM, and GraphSAM to verify the role of each module. In summary, we have the following two observations.

▷ The size of $\rho$ directly impacts the generalization of SAM and GraphSAM for different datasets. In particular, we investigate the impact of different fixed-value $\rho$ and gradient ball's size $\rho$ schedulers on the performance of various datasets. We list their performances in Table 8, and conduct that: (1) The $\rho$ of the gradient ball helps SAM and GraphSAM to seek the model parameters that lie in neighborhoods having uniformly low loss. We need to manually adjust the size of $\rho$ to get better performance for different datasets (e.g., the best performance of BBBP is $\rho = 0.05$ but ESOL is $\rho = 0.001$ in SAM). Due to the nature of GraphSAM, which needs a small $\rho$ to approximate the ground-truth, $\rho$ is unsuitable for over-scaling. (2) Compared to fixed $\rho$, the scheduler can improve the model's generalization performance on different datasets. The reason for the above conduct is that parameter re-scaling can cause a difference in sharpness values so the size of $\rho$ may weaken the correlation between sharpness and generalization gap (Dinh et al., 2017), this phenomenon is named the scale-dependency problem. Our gradient ball's size $\rho$ scheduler module, which changes the size of rho periodically, can remedy the scale-dependency problem of sharpness. Experimentally, GraphSAM and SAM have relatively stable generalization performance in different datasets under the action of the scheduler.

▷ Accuracy-efficiency trade-off. The secret of GraphSAM's ability to maintain similar performance to SAM is the timely re-anchor of $\epsilon_t$. The effectiveness of another module of GraphSAM, moving average, has been shown in Table 1. In this part, we mainly verify the correlation between accuracy-efficiency. We propose the GraphSAM-$K$ to investigate the impact of the rate of re-anchor to the perturbation gradient $\epsilon_t$ on generalization performance. The $K$ means that we perform an additional forward and backward propagation to re-anchor from $\epsilon_t$ to $\epsilon_0$ for every $K$ epoch. As in Fig. 6, we analyze GraphSAM-$K$ for different values of $K$. The $N$ means that only once re-anchor of $\epsilon_t$. When $K = 1$, GraphSAM is comparable to SAM. When $K = 2$, the performance of GraphSAM is slightly higher than Adam's. When $K > 2$, the efficiency of GraphSAM is increasing rapidly, but its performance drops sharply. The reason is that as the training steps increase, the error between $\hat{\epsilon}_t$ obtained from equation 3 and the ground-truth $\hat{\epsilon}_S$ becomes increasingly large. To reduce the error, we need to re-anchor the $\epsilon_t$ by forward and backward propagation in each epoch repeatedly.

## 6 CONCLUSION

In this paper, we perform a series of analytical experiments to show that the graph transformer model suffers from sharp local minima when the pre-training process is removed. SAM can solve this problem, but its computational loss is double that of the traditional optimizer. Then we propose an efficient algorithm GraphSAM which reduces the training cost of SAM and improves the generalization performance of graph transformer models. The experiments show the superiority of GraphSAM, especially in optimizing the model update process.

# 7 ACKNOWLEDGMENTS

This work was supported by a grant from the National Natural Science Foundation of China under grants (No.62372211, 62272191), the Foundation of the National Key Research and Development of China (No.2021ZD0112500), and the International Science and Technology Cooperation Program of Jilin Province (No.20230402076GH), and the Science and Technology Development Program of Jilin Province (No. 20220201153GX).

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

# A APPENDIX

## A.1 APPENDIX 1: PROOF FOR CONJECTURE 2

**Conjecture 2.** Let $\parallel \mathcal{L}_{\mathbb{G}}(\theta + \hat{\epsilon}_{\mathrm{G}}) - \mathcal{L}_{\mathbb{G}}(\theta + \hat{\epsilon}_{\mathrm{S}}) \parallel$ denote the gap of generalization performance between SAM and GraphSAM. It is positively correlated with the gradient approximation error as below:

$$\parallel \mathcal{L}_{\mathbb{G}}(\theta + \hat{\epsilon}_{\mathrm{G}}) - \mathcal{L}_{\mathbb{G}}(\theta + \hat{\epsilon}_{\mathrm{S}}) \parallel \propto \parallel \hat{\epsilon}_{\mathrm{G}} - \hat{\epsilon}_{\mathrm{S}} \parallel .$$

**Proof 1.** For SAM loss function $\mathcal{L}_{\mathbb{G}}(\theta + \hat{\epsilon})$, based on Taylor Expansion (Liu et al., 2022), we can obtain:

$$\mathcal{L}_{\mathbb{G}}(\theta + \hat{\epsilon}) \approx \mathcal{L}_{\mathbb{G}}(\theta) + \hat{\epsilon}\nabla_\theta \mathcal{L}_{\mathbb{G}}(\theta) \tag{6}$$

Therefore, we have:

$$\begin{aligned}
&\mathcal{L}_{\mathbb{G}}(\theta + \hat{\epsilon}_{\mathrm{G}}) - \mathcal{L}_{\mathbb{G}}(\theta + \hat{\epsilon}_{\mathrm{S}}) \\
&= (\mathcal{L}_{\mathbb{G}} + \hat{\epsilon}_{\mathrm{G}}\nabla_\theta \mathcal{L}_{\mathbb{G}}(\theta)) - (\mathcal{L}_{\mathbb{G}} + \hat{\epsilon}_{\mathrm{S}}\nabla_\theta \mathcal{L}_{\mathbb{G}}(\theta)) \\
&= (\hat{\epsilon}_{\mathrm{G}} - \hat{\epsilon}_{\mathrm{S}})\nabla_\theta \mathcal{L}_{\mathbb{G}}(\theta)
\end{aligned} \tag{7}$$

According to equation 7, the gap in generalization performance between SAM and GraphSAM is proportional to the gradient approximation error.

## A.2 APPENDIX 2: PROOF FOR CONJECTURE 1

**Conjecture 1.** *Let $\hat{\epsilon}_{\mathrm{S}}$ and $\hat{\epsilon}_{\mathrm{G}}$ be the projected perturbation gradients of SAM and GraphSAM, respectively. We ignore the subscript t for the simple representation. For any $\rho > 0$, we assume that $||\hat{\epsilon}_{\mathrm{S}}||_2 < ||\hat{\epsilon}_{\mathrm{G}}||_2$ and $\hat{\epsilon}_{\mathrm{S}}$ is the ground-truth, we have*

$$\max_{||\hat{\epsilon}_{\mathrm{S}}||_2 \le \rho} \mathcal{L}_{\mathbb{G}}(\theta + \hat{\epsilon}_{\mathrm{S}}) \le \max_{||\hat{\epsilon}_{\mathrm{G}}||_2 \le \rho} \mathcal{L}_{\mathbb{G}}(\theta + \hat{\epsilon}_{\mathrm{G}}). \tag{8}$$

**Proof 2.** For SAM loss function Combining equation 2 of the main paper, we can see that the size of $\hat{\epsilon}_t$ is determined by $\epsilon_t$. Then $\epsilon_t^{\mathrm{G}}$ is expressed as follows:

$$\epsilon_t^{\mathrm{G}} = \beta\epsilon_{t-1}^{\mathrm{G}} + (1 - \beta)\frac{\omega_{t-1}}{||\omega_{t-1}||_2}. \tag{9}$$

(1) When $t = 0$, $\epsilon_t^{\mathrm{G}} = \epsilon_t^{\mathrm{S}}$, because both of them are computed by the same forward and backward propagation.

(2) When $t = 1$, $\epsilon_t^{\mathrm{G}} = \beta\epsilon_0^{\mathrm{G}} + (1 - \beta)\frac{\omega_0}{||\omega_0||_2} > \epsilon_t^{\mathrm{S}}$,

$$\begin{aligned}
\epsilon_t^{\mathrm{G}} - \epsilon_t^{\mathrm{S}} &= \beta(\epsilon_0^{\mathrm{G}} - \epsilon_1^{\mathrm{S}}) + (1 - \beta)(\frac{\omega_0}{||\omega_0||_2} - \epsilon_1^{\mathrm{S}}) \\
&= \beta(\epsilon_0^{\mathrm{S}} - \epsilon_1^{\mathrm{S}}) + (1 - \beta)(\frac{\omega_0}{||\omega_0||_2} - \epsilon_1^{\mathrm{S}}),
\end{aligned} \tag{10}$$

as shown in Fig. 3(a) of the main paper, the change of perturbation gradient is small, and after experimental analysis, the updating gradient $\omega_t \gg \epsilon_t$. So $\epsilon_t^{\mathrm{G}} > \epsilon_t^{\mathrm{S}}$, for $t = 1$.

(3) When $t = 2$, because of $\epsilon_1^{\mathrm{G}} > \epsilon_1^{\mathrm{S}}$, we have

$$\begin{aligned}
\epsilon_t^{\mathrm{G}} - \epsilon_t^{\mathrm{S}} &= \beta(\epsilon_1^{\mathrm{G}} - \epsilon_2^{\mathrm{S}}) + (1 - \beta)(\frac{\omega_1}{||\omega_1||_2} - \epsilon_2^{\mathrm{S}}) \\
&> \beta(\epsilon_1^{\mathrm{S}} - \epsilon_2^{\mathrm{S}}) + (1 - \beta)(\frac{\omega_1}{||\omega_{t-1}||_2} - \epsilon_2^{\mathrm{S}}),
\end{aligned}$$

so $\epsilon_t^{\mathrm{G}} > \epsilon_t^{\mathrm{S}}$, for $t = 2$.

(4) When $t > 2$, analogously to the above conclusions, we have

$$\epsilon_t^{\mathrm{G}} - \epsilon_t^{\mathrm{S}} = \beta(\epsilon_{t-1}^{\mathrm{G}} - \epsilon_t^{\mathrm{S}}) + (1 - \beta)(\frac{\omega_{t-1}}{||\omega_{t-1}||_2} - \epsilon_2^{\mathrm{S}})$$

$$> \beta(\epsilon_{t-1}^{\mathrm{S}} - \epsilon_t^{\mathrm{S}}) + (1 - \beta)(\frac{\omega_{t-1}}{||\omega_{t-1}||_2} - \epsilon_t^{\mathrm{S}})$$

In summary, we can conclude that $\epsilon_t^{\mathrm{G}} > \epsilon_t^{\mathrm{S}}$, for $t > 0$.

Following the above, the equation 7 can be rewritten as follows:

$$\mathcal{L}_{\mathbb{G}}(\theta_t + \hat{\epsilon}_t^{\mathrm{G}}) - \mathcal{L}_{\mathbb{G}}(\theta_t + \hat{\epsilon}_t^{\mathrm{S}})$$

$$\approx (\hat{\epsilon}_t^{\mathrm{G}} - \hat{\epsilon}_t^{\mathrm{S}})\nabla_\theta \mathcal{L}_{\mathbb{G}}(\theta_t) \tag{11}$$

$$\approx (\epsilon_t^{\mathrm{G}} - \epsilon_t^{\mathrm{S}})\nabla_\theta \mathcal{L}_{\mathbb{G}}(\theta_t) > 0.$$

Therefore, it can be proved that

$$\max_{||\hat{\epsilon}_{\mathrm{S}}||_2 \leq \rho} \mathcal{L}_{\mathbb{G}}(\theta + \hat{\epsilon}_{\mathrm{S}}) < \max_{||\hat{\epsilon}_{\mathrm{G}}||_2 \leq \rho} \mathcal{L}_{\mathbb{G}}(\theta + \hat{\epsilon}_{\mathrm{G}})$$

Also when $\rho = 0$ or $t = 0$, then

$$\max_{||\hat{\epsilon}_{\mathrm{S}}||_2 \leq \rho} \mathcal{L}_{\mathbb{G}}(\theta + \hat{\epsilon}_{\mathrm{S}}) = \max_{||\hat{\epsilon}_{\mathrm{G}}||_2 \leq \rho} \mathcal{L}_{\mathbb{G}}(\theta + \hat{\epsilon}_{\mathrm{G}})$$

Although we prove the correctness of $Theorem$ 1 equation 8 by *Mathematical Induction* and experiment, we still want to find out the main factors affecting the perturbation gradient of GraphSAM, so we expand equation 9 as follows:

$$\epsilon_t^{\mathrm{G}} = \beta\epsilon_{t-1}^{\mathrm{G}} + (1 - \beta)\frac{\omega_{t-1}}{||\omega_{t-1}||_2}$$

$$= \beta[\beta\epsilon_{t-2}^{\mathrm{G}} + (1 - \beta)\frac{\omega_{t-2}}{||\omega_{t-2}||_2}] + (1 - \beta)\frac{\omega_{t-1}}{||\omega_{t-1}||_2}$$

$$= \beta^2\epsilon_{t-2}^{\mathrm{G}} + \beta(1 - \beta)\frac{\omega_{t-2}}{||\omega_{t-2}||_2} + (1 - \beta)\frac{\omega_{t-1}}{||\omega_{t-1}||_2} \tag{12}$$

$$\vdots$$

$$= \beta^t\epsilon_0^{\mathrm{G}} + \beta^{t-1}(1 - \beta)\frac{\omega_0}{||\omega_0||_2} + \cdots + \beta^0(1 - \beta)\frac{\omega_{t-1}}{||\omega_{t-1}||_2}.$$

As equation 12, we find that the parameter $\beta$ of the moving average plays a key role. So we perform some experiments on it. As shown in Table 3, it is known experimentally that GraphSAM performs best when $\beta = 0.99$. This result is particularly evident in the ESOL dataset. So equation 10 holds for a suitable $\beta$. In the paper, we experimentally observe that the perturbation gradient and the updating gradient differ by two orders of magnitude, i.e., 100 times. Therefore, the optimal result is $\beta = 0.99$, so $1 - \beta$=0.01.

Table 3: The influence of smoothing parameter $\beta$ of moving average on GraphSAM algorithm on CoMPT.

| Algorithm | $\beta$ | BBBP (ROC-AUC↑) | ClinTox(ROC-AUC↑) | ESOL (RMSE↓) | Lipophilicity (RMSE↓) |
|---|---|---|---|---|---|
| | 0.9 | $0.958 \pm 0.014$ | $0.928 \pm 0.015$ | $0.557 \pm 0.020$ | $0.614 \pm 0.021$ |
| GraphSAM | 0.99 | $\mathbf{0.961 \pm 0.012}$ | $\mathbf{0.937 \pm 0.008}$ | $\mathbf{0.511 \pm 0.018}$ | $\mathbf{0.608 \pm 0.007}$ |
| | 0.999 | $0.959 \pm 0.003$ | $0.931 \pm 0.006$ | $0.549 \pm 0.023$ | $0.612 \pm 0.010$ |

A.3 Appendix 3: The Statistics of Datasets

GraphSAM and SAM is evaluated on six molecular graph datasets, as described below:

**Molecular Classification Datasets.**

- **BBBP (Martins et al., 2012):** This is a Blood-brain barrier penetration (BBBP) dataset, which involves recording whether a compound has the permeability to penetrate the blood-brain barrier. It has binary labels for 2,039 molecules.

- **Tox21 (Capuzzi et al., 2016):** This is a public database for measuring the toxicy of compounds, which contains 7,831 compounds against 12 different targets.

- **Sider (Kuhn et al., 2016):** The Side Effect Resource (Sider) is a database of marketed drugs and adverse drug reactions, which contains 1,427 approved drugs with 27 system organ branches.

- **Clintox (Gayvert et al., 2016):** It has 1,478 drugs and compares them approved through FDA and drugs eliminated due to the toxicity during clinical trials.

**Molecular Regression Datasets.**

- **ESOL (Delaney, 2004):** It is a small dataset with 1,128 molecules to document the solubility of compounds.

- **Lipophilicity (Gaulton et al., 2012).** it was selected from the ChEMBL database, and contains 4,198 molecules to determine the lipophilicity of the molecule.

The statistics of the molecular graph datasets used in the graph classification and graph regression are summarized in Table 4.

Table 4: Statistics of datasets.

| Category | Dataset | #Tasks | Task Type | #Molecule |
|---|---|---|---|---|
| Physiology | BBBP | 1 | Classification | 2,039 |
| | Tox21 | 12 | Classification | 7,831 |
| | Sider | 27 | Classification | 1,427 |
| | Clintox | 2 | Classification | 1,478 |
| Physical chemistry | ESOL | 1 | Regression | 1,128 |
| | Lipophilicity | 1 | Regression | 4,198 |

In addition to the above datasets, we have performed experiments on three additional datasets from biophysics (BACE) and quantum mechanics (QM7, QM8, QM9), as shown in Table 5 and Table 6. This demonstrates the generalization capability of GraphSAM.

**Quantum Mechanics Datasets.**

- **QM7 (Blum & Reymond, 2009):** The QM7 dataset is a subset of the GDB-13 database, which contains 7615 stable, synthesizable organic molecules.

- **QM8 (Ramakrishnan et al., 2015):** The QM8 dataset comes from a quantum mechanical computational modeling study of the electronic energy spectra and excited state energies of small molecules. It contains 21786 samples.

**Biophysics Dataset.**

- **BACESubramanian et al. (2016):** The BACE dataset provides combined quantitative (IC50) and qualitative (binary labeling) results for a set of human $\beta$-secretase 1 (BACE-1) inhibitors and contains data for 1,522 molecules. All data are experimental values reported in the scientific literature over the last decade, some of which also provide detailed crystal structures.

Table 5: Statistics of three additional datasets.

| Category | Dataset | #Tasks | Task Type | #Molecule |
|---|---|---|---|---|
| Quantum mechanics | QM7 | 1 | Regression | 7,165 |
| | QM8 | 12 | Regression | 21,786 |
| Biophysics | BACE | 1 | Classification | 1,522 |

Table 6: Prediction results of GraphSAM and baselines.

| | BACE | QM7 | QM8 |
|---|---|---|---|
| **GROVER** | ROC-AUC ↑ | MAE ↓ | MAE ↓ |
| Adam | 0.871 | 78.9 | 0.0203 |
| SAM | **0.886** | 76.4 | 0.0189 |
| GraphSAM | 0.882 | **75.8** | **0.0185** |
| **CoMPT** | ROC-AUC ↑ | MAE ↓ | MAE ↓ |
| Adam | 0.863 | 66.1 | 0.0159 |
| SAM | 0.876 | 64.3 | 0.0145 |
| GraphSAM | **0.880** | **64.1** | **0.0141** |

## A.4 APPENDIX 4: TRAINING DETAILS AND ALGORITHM DETAILS

**Backbone frameworks and baselines.** To implement SAM and GraphSAM on molecular graph data with two widely-used graph transformer backbones, GROVER (Rong et al., 2020) and CoMPT (Chen et al., 2021a). We comprehensively compare them against four baselines in the graph-level task. GCN (Kipf & Welling, 2016) is the most classical graph convolutional neural network. Compared with GCN, MPNN (Gilmer et al., 2017) and its two variants (DMPNN (Yang et al., 2019) and CMPNN (Song et al., 2020)) integrate the edge features into the message-passing process.

**SAM and Efficient Variants of SAM.** Here we analyze the advantages and disadvantages of SAM and the efficient variants of SAM, and verify them through experiments. As shown in Table 7.

- **SAM (Foret et al., 2020):** First work on Sharpness-Aware Minimization(SAM) type methods. SAM computes the perturbation gradient and the updating gradient by two forward propagation and backpropagation, respectively. This prevents the model parameters from falling into sharp local minima. The disadvantage is twice the computational loss of the base optimizer.

- **LookSAM (Liu et al., 2022):** An efficient approach to SAM variants. It saves computational loss by observing the training gradient of the CV domain model and periodically computing two gradients. The disadvantage is that it only cares about the gradient direction but ignores the gradient size. This leads to poor results when applied to other domains if the gradient direction is not similar to the CV domain. When the $\rho$ is tuned down, the performance of LookSAM is only close to the base optimizer.

- **AE-SAM (Jiang et al., 2023):** A generic hot-swappable component AE-SAM utilizes Squared stochastic gradient norms to decide whether to compute only the perturbation gradient and not the updating gradient. When Squared stochastic gradient norms exceed a certain threshold, quadratic gradient computation is performed and vice versa. The trend of its Squared stochastic gradient norms in the CV domain is not the same as in the molecular graph domain. Therefore its performance varies greatly when encountering different models and data while ignoring gradient direction and gradient size.

- **RST (Zhao et al., 2022a):** A method that utilizes the Bernoulli trial to periodically compute two gradients. It is more stochastic and faster to calculate than AE-SAM. However, its drawbacks are also apparent, aimlessly deciding whether to update the gradient or not ends up with poorer performance. Its performance is only slightly higher than SAM-One and lower than SAM-k.

- **GraphSAM:** An efficient SAM-type method based on gradient descent trend specifically designed for transformer models in the molecular graph domain. In contrast to the single improvement of other SAM-type methods, we design the gradient approximation method for the gradient direction. We design an adjustable gradient ball's size ($\rho$) scheduler for the gradient size. In addition, we

have designed a method to compute the gradient periodically so that GraphSAM maintains a high level of performance improvement. Last but not least, we prove the effectiveness of GraphSAM through experimentation and derivation.

**Implementations.** The backbone implementations and their hyperparameter settings are provided by the publicly released repositoriesRong et al. (2020); Chen et al. (2021a). GROVER and CoMPT utilize Adam as the base optimizer, and neither uses the pre-training strategy. We only adjust the specific hyperparameters introduced by GraphSAM: (1) smoothing parameters of moving average $\beta$ is tuned within {0.9, 0.99, 0.999}, (2) the initial size of the gradient ball $\rho$ is selected from {0.05, 0.01, 0.005, 0.001}, (3) the $\rho$'s update rate $\lambda$ is searched over {1, 3, 5}, (4) and the scheduler's modification scale $\gamma = \{0.5, 0.2, 0.1\}$. Due to space limitations, we place our experiments on hyperparameters in Appendix A.6. All the experiments are implemented by PyTorch, and run on an NVIDIA TITAN-RTX (24G) GPU.

GraphSAM uses uniform hyperparameters for different models[4][5] and datasets to train with the following settings: (1) $initial\_\rho = 0.05$; (2) $\gamma = 0.5$; (3) $\beta = 0.99$; (4) $\lambda = 1$. For the other model parameters, we use the model parameter settings of the original paper. LookSAM uses the hyperparameters ($\rho = 0.0001$, $\alpha = 0.2$, k = 8) for training. SAM-k uses k=8, where k stands for step instead of epoch. And GraphSAM algorithm is shown in Algorithm 1.

---

**Algorithm 1** GraphSAM

---

**Require:** Network $f_\theta$; Training set $\mathbb{G}$; Batch size $b$; Neighborhood size $\rho > 0$; Moving average hyperparameter $\beta$; Learning rate $\eta > 0$; $\rho$'s update rate $\lambda > 0$; The modification scale $\gamma$.
 1: **for** $epoch \longleftarrow 0,1,2,\cdots E$ **do**
 2:     Sample a mini-batch in $\mathbb{B} \subset \mathbb{G}$ with size b.
 3:     **for** $t \longleftarrow 0,1,2,\cdots N$ **do**
 4:       **if** t == 0: **then**
 5:         $\epsilon_t = \nabla_\theta \mathcal{L}_{\mathbb{B}}(\theta_t)$
 6:         **if** $epoch\%\lambda == 0$: **then**
 7:           $\rho_{\text{new}} = \rho_{\text{initial}} * \gamma^{\text{epoch}/\lambda}$
 8:         **end if**
 9:       **else**
10:         $\epsilon_t = \beta\epsilon_{t-1} + (1-\beta)\omega_{t-1}/ \parallel \omega_{t-1} \parallel_2$
11:       **end if**
12:       $\hat{\epsilon}_t = \rho \cdot \text{sign}(\epsilon_t)\frac{|\epsilon_t|}{||\epsilon_t||_2}$
13:       $\omega_t = \nabla_\theta \mathcal{L}_{\mathbb{B}}(\theta_t)|_{\theta_t + \hat{\epsilon}_t}$
14:       $\theta_{t+1} = \theta_t - \eta \cdot \omega_t$
15:     **end for**
16: **end for**
17: **Return:** $\theta_t$
**Ensure:** Model trained with GraphSAM.

---

[4]https://github.com/tencent-ailab/GROVER/
[5]https://github.com/jcchan23/CoMPT/

## A.5 APPENDIX 5: SUPPLEMENTARY EXPERIMENTS.

In this section, we mainly add some experiments to sections 5.2.2 (Throughput and accuracy of the optimizers) and 5.2.3 (Roles of GraphSAM modules) of the main paper.

①We compare the throughput and generalization performance of GraphSAM with other optimization algorithms in Table 7. Where throughput denotes the computational overhead which is quantified by graphs processed per second (Graphs/s), and the generalization performance is measured by test ROC-AUC/RMSE.

We can see that AE-SAM is more stable compared to RST and LookSAM. This depends on how high its throughput is, i.e., whether it computes the two gradients multiple times. When its throughput is high, its performance will be relatively lower, and vice versa. GraphSAM, on the other hand, performs very consistently, improving both the throughput and the generalization performance of the model, which is attributed to the three improvement parts of this paper. In addition, due to LookSAM and RST's own shortcomings for molecular graphs, their performance improvement is not sufficient and may even be counterproductive. For example, RST is only close to the simple SAM-k method in most cases.

Table 7: Classification accuracy and training speed. The numbers in parentheses (·) indicate the ratio of GraphSAM's training speed w.r.t. SAM.

| | Tox21 | | Sider | | ClinTox | |
|---|---|---|---|---|---|---|
| **GROVER** | ROC-AUC↑ | Throughput | ROC-AUC↑ | Throughput | ROC-AUC↑ | Throughput |
| Adam | 0.822 | 131 | 0.649 | 130 | 0.853 | 161 |
| SAM | 0.840 | 81(100.0%) | 0.660 | 79(100.0%) | **0.872** | 110(100.0%) |
| SAM-One | 0.801 | 118(145.6%) | 0.610 | 114(144.3%) | 0.818 | 150(136.4%) |
| SAM-$k$ | 0.815 | 109(134.6%) | 0.631 | 105(132.9%) | 0.836 | 141(128.1%) |
| LookSAM | 0.829 | 92(113.6%) | 0.653 | 95(120.3%) | 0.847 | 128(116.4%) |
| AE-SAM | 0.834 | 90(111.1%) | 0.649 | 101(127.8%) | 0.858 | 133(121.0%) |
| RST | 0.818 | 100(123.4%) | 0.625 | 103(130.3%) | 0.845 | 143(130.0%) |
| GraphSAM | **0.846** | 98(121.0%) | **0.665** | 98(124.1%) | 0.866 | 134(121.8%) |
| **CoMPT** | ROC-AUC↑ | Throughput | ROC-AUC↑ | Throughput | ROC-AUC↑ | Throughput |
| Adam | 0.828 | 119 | 0.621 | 219 | 0.914 | 211 |
| SAM | 0.839 | 60(100.0%) | 0.643 | 114(100.0%) | 0.927 | 115(100.0%) |
| SAM-One | 0.788 | 102(170.0%) | 0.595 | 198(173.6%) | 0.879 | 201(174.7%) |
| SAM-$k$ | 0.819 | 94(156.6%) | 0.610 | 180(157.8%) | 0.895 | 187(162.6%) |
| LookSAM | 0.825 | 78(130.0%) | 0.625 | 141(123.6%) | 0.916 | 147(127.8%) |
| AE-SAM | 0.833 | 71(118.3%) | 0.631 | 150(131.6%) | 0.909 | 155(134.8%) |
| RST | 0.815 | 94(156.6%) | 0.615 | 169(148.2%) | 0.890 | 180(156.5%) |
| GraphSAM | **0.841** | 85(141.7%) | **0.645** | 153(134.2%) | **0.937** | 160(139.1%) |

| | ESOL | | Lipophilicity | |
|---|---|---|---|---|
| **GROVER** | RMSE↓ | Throughput | RMSE↓ | Throughput |
| Adam | 0.639 | 360 | 0.671 | 160 |
| SAM | **0.619** | 223(100.0%) | 0.662 | 68(100.0%) |
| SAM-One | 0.671 | 332(148.9%) | 0.715 | 143(210.3%) |
| SAM-$k$ | 0.650 | 298(133.6%) | 0.698 | 129(189.7%) |
| LookSAM | 0.633 | 278(126.1%) | 0.680 | 90(132.4%) |
| AE-SAM | 0.631 | 289(129.6%) | 0.674 | 100(147.0%) |
| RST | 0.638 | 305(130.9%) | 0.690 | 116(170.1%) |
| GraphSAM | 0.625 | 282(126.5%) | **0.654** | 95(139.7%) |
| **CoMPT** | RMSE↓ | Throughput | RMSE↓ | Throughput |
| Adam | 0.562 | 502 | 0.618 | 432 |
| SAM | 0.517 | 282(100.0%) | 0.611 | 234(100.0%) |
| SAM-One | 0.601 | 457(162.1%) | 0.638 | 392(167.5%) |
| SAM-$k$ | 0.575 | 413(146.5%) | 0.630 | 367(156.8%) |
| LookSAM | 0.543 | 338(119.9%) | 0.621 | 318(135.9%) |
| AE-SAM | 0.527 | 329(116.7%) | 0.615 | 308(131.6%) |
| RST | 0.588 | 391(138.6%) | 0.633 | 371(158.5%) |
| GraphSAM | **0.511** | 347(123.0%) | **0.608** | 331(141.5%) |

②We investigate the impact of different fixed-value $\rho$ and gradient ball's size $\rho$ schedulers on the performance of various datasets. We list their performances in Table 8.

We find that our proposed Gradient ball's size $(\rho)$ schedule is more stable than the fixed $(\rho)$, and eliminates the need for extensive tuning to find the optimal rho.

Table 8: The influence of gradient ball's size $\rho$ on SAM and GraphSAM on the CoMPT.

| Algorithm | $\rho$ | BBBP(ROC-AUC↑) | Tox21(ROC-AUC↑) | Sider(ROC-AUC↑) |
|---|---|---|---|---|
| **SAM** | 0.05 | $\underline{0.957 \pm 0.010}$ | $0.822 \pm 0.011$ | $0.621 \pm 0.018$ |
| | 0.005 | $0.952 \pm 0.022$ | $\underline{0.842 \pm 0.008}$ | $0.635 \pm 0.021$ |
| | 0.001 | $0.953 \pm 0.021$ | $0.835 \pm 0.013$ | $\underline{0.641 \pm 0.029}$ |
| | scheduler | $\mathbf{0.962 \pm 0.033}$ | $\mathbf{0.839 \pm 0.006}$ | $\mathbf{0.643 \pm 0.049}$ |
| **GraphSAM** | 0.05 | $0.933 \pm 0.025$ | $0.810 \pm 0.033$ | $0.618 \pm 0.029$ |
| | 0.005 | $0.949 \pm 0.020$ | $0.826 \pm 0.020$ | $0.625 \pm 0.018$ |
| | 0.001 | $\mathbf{0.964 \pm 0.008}$ | $\underline{0.838 \pm 0.006}$ | $\underline{0.636 \pm 0.016}$ |
| | scheduler | $\underline{0.961 \pm 0.012}$ | $\mathbf{0.841 \pm 0.004}$ | $\mathbf{0.645 \pm 0.013}$ |

| Algorithm | $\rho$ | ClinTox(ROC-AUC↑) | ESOL(RMSE↓) | Lipophilicity(RMSE↓) |
|---|---|---|---|---|
| **SAM** | 0.05 | $0.910 \pm 0.031$ | $0.539 \pm 0.031$ | $0.622 \pm 0.016$ |
| | 0.005 | $0.918 \pm 0.016$ | $0.534 \pm 0.021$ | $\mathbf{0.601 \pm 0.009}$ |
| | 0.001 | $\underline{0.924 \pm 0.018}$ | $\underline{0.527 \pm 0.014}$ | $0.613 \pm 0.018$ |
| | scheduler | $\mathbf{0.927 \pm 0.025}$ | $\mathbf{0.517 \pm 0.025}$ | $\underline{0.611 \pm 0.015}$ |
| **GraphSAM** | 0.05 | $0.902 \pm 0.045$ | $0.946 \pm 0.103$ | $0.691 \pm 0.053$ |
| | 0.005 | $0.924 \pm 0.020$ | $0.598 \pm 0.212$ | $0.625 \pm 0.018$ |
| | 0.001 | $\underline{0.931 \pm 0.008}$ | $\underline{0.528 \pm 0.016}$ | $\underline{0.610 \pm 0.016}$ |
| | scheduler | $\mathbf{0.937 \pm 0.008}$ | $\mathbf{0.511 \pm 0.014}$ | $\mathbf{0.608 \pm 0.007}$ |

▷ Accuracy-efficiency trade-off. The secret of GraphSAM's ability to maintain similar performance to SAM is the timely re-anchor of $\epsilon_t$. The effectiveness of another module of GraphSAM, moving average, has been shown in Table 1. In this part, we mainly verify the correlation between accuracy-efficiency. We propose the GraphSAM-$K$ to investigate the impact of the rate of re-anchor to the perturbation gradient $\epsilon_t$ on generalization performance. The $K$ means that we perform an additional forward and backward propagation to re-anchor from $\epsilon_t$ to $\epsilon_0$ for every $K$ epoch. As in Fig. 6, we analyze GraphSAM-$K$ for different values of $K$. The $N$ means that only once re-anchor of $\epsilon_t$. When $K = 1$, GraphSAM is comparable to SAM. When $K = 2$, the performance of GraphSAM is slightly higher than Adam's. When $K > 2$, the efficiency of GraphSAM is increasing rapidly, but its performance drops sharply. The reason is that as the training steps increase, the error between $\hat{\epsilon}_t$ obtained from equation 3 and the ground-truth $\hat{\epsilon}_S$ becomes increasingly large. To reduce the error, we need to re-anchor the $\epsilon_t$ by forward and backward propagation in each epoch repeatedly.

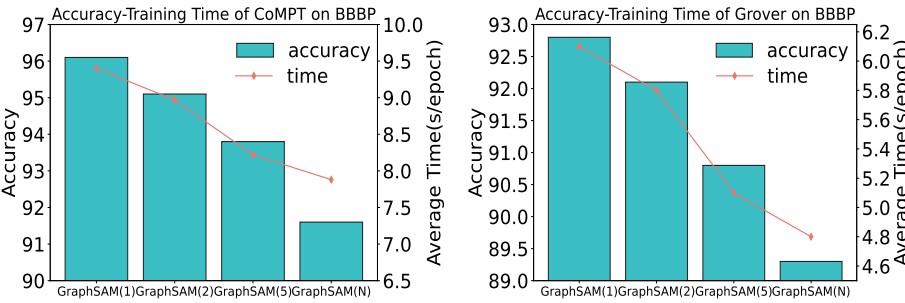

Figure 6: Accuracy-Training Time of different models for GraphSAM-$K$. Average Time (s/epoch) represents the average time consumed for each epoch.

A.6    APPENDIX 6: HYPERPARAMETER STUDIES.

We conduct experiments on some hyperparameters introduced by GraphSAM in the training process.

❻ *The appropriate size of $\rho$ in GraphSAM is essential for model training.* We conduct the experiment and observe that the size of $\rho$ is related to the initial size $\rho$, $\rho$'s update rate $\lambda$, and the scheduler's modification scale $\gamma$. To effectively evaluate the impact of the size of $\rho$, the default values of the hyperparameters in this paper are initial $\rho = 0.05$, $\lambda = 1$, and $\gamma = 0.5$. In Fig. 7a, we find that if the initial $\rho$ is small, the weight perturbation has a smaller range, which eventually affects the training performance of GraphSAM. In addition, if the initial $\rho = 0.05$ is large and the update rate $\lambda$ is big (i.e., $\rho$ is updated more slowly), the perturbation range of GraphSAM deviates severely from the SAM, causing the model generalization performance to drop sharply as shown in Fig. 7b. Similarly, when the modification scale of $\rho$ is faster i.e. $\gamma = 0.1$, the weight perturbation range spans a larger extent, making the performance of GraphSAM scaled down, especially in the ESOL dataset in Table 9. When the modification scale of $\rho$ is faster i.e. $\gamma = 0.1$, the weight perturbation range spans a larger extent, making the performance of GraphSAM scaled down, especially in the ESOL dataset in Table 9.

Table 9: The influence of the gradient ball's size $\rho$ scheduler's modification scale $\gamma$ on GraphSAM algorithm with CoMPT.

| Algorithm | $\gamma$ | BBBP (ROC-AUC↑) | ClinTox(ROC-AUC↑) | ESOL (RMSE↓) | Lipophilicity (RMSE↓) |
|---|---|---|---|---|---|
| | 0.5 | **0.961 ± 0.012** | **0.937 ± 0.008** | **0.511 ± 0.018** | **0.608 ± 0.007** |
| GraphSAM | 0.2 | 0.959 ± 0.002 | 0.923 ± 0.015 | 0.560 ± 0.010 | 0.618 ± 0.019 |
| | 0.1 | 0.956 ± 0.005 | 0.911 ± 0.011 | 0.608 ± 0.012 | 0.634 ± 0.010 |

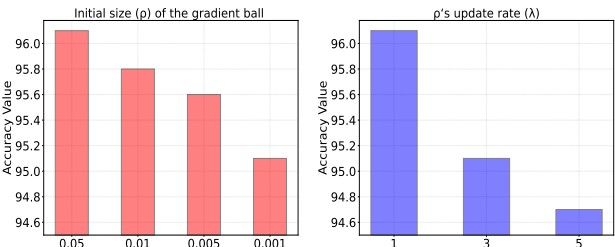

Figure 7: Hyperparameter studies of $\rho$, and $\lambda$ of CoMPT+ GraphSAM on BBBP dataset.

A.7    APPENDIX 7: SUPPLEMENT TO THE PRE-TRAINING EXPERIMENT.

We have also conducted experiments on the pre-training method, and both SAM and GraphSAM have certain performance improvements. However, compared to non-pre-training experiments, the improvement is limited. As shown in table 10, we provide the experimental results of SAM and GraphSAM on the pre-trained model GROVER with six datasets.

Table 10: Prediction results of GraphSAM on pretraining-GROVER on six datasets. We used 5-fold cross-validation with random split and replicated experiments on each task five times. The mean and standard deviation of AUC or RMSE values are reported.

| Task | Graph Classification (ROC-AUC↑) | | | | Graph Regression (RMSE↓) | |
|---|---|---|---|---|---|---|
| Dataset | BBBP | Tox21 | Sider | ClinTox | ESOL | Lipophilicity |
| GROVER | $0.930_{\pm 0.016}$ | $0.833_{\pm 0.028}$ | $0.661_{\pm 0.030}$ | $0.931_{\pm 0.011}$ | $0.627_{\pm 0.087}$ | $0.544_{\pm 0.039}$ |
| + SAM | $0.935_{\pm 0.025}$ | $0.836_{\pm 0.021}$ | $0.669_{\pm 0.028}$ | $\mathbf{0.940_{\pm 0.018}}$ | $0.623_{\pm 0.103}$ | $0.537_{\pm 0.033}$ |
| + GraphSAM | $\mathbf{0.939_{\pm 0.011}}$ | $\mathbf{0.838_{\pm 0.033}}$ | $\mathbf{0.673_{\pm 0.036}}$ | $0.938_{\pm 0.020}$ | $\mathbf{0.619_{\pm 0.111}}$ | $\mathbf{0.529_{\pm 0.030}}$ |

A.8    APPENDIX 8: ADDITIONS TO THE OBSERVATION OF GRADIENT VARIATION.

In this section, we address the phenomenon detailed as Observation I in the main text, as illustrated in Fig. 8. This addition robustly affirms that the variations observed in the perturbation gradient and the updating gradient in Observation I are not isolated instances, but rather a general trend. Furthermore, we incorporate the trajectory of changes in the magnitudes of $||\epsilon_t||_2$ and $||\omega_t||_2$ as depicted in Fig. 9. This inclusion substantiates our finding $||\omega_t||_2 \gg ||\epsilon_t||_2$ , confirming the initial observation and providing deeper insight into the underlying dynamics at play.

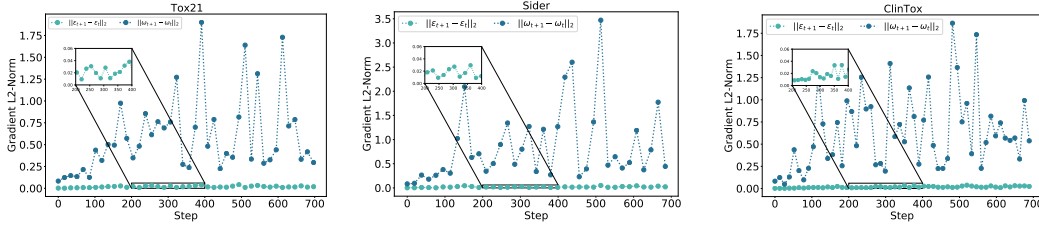

Figure 8: Illustration on the observation of gradient variation during training on GROVER with three datasets.

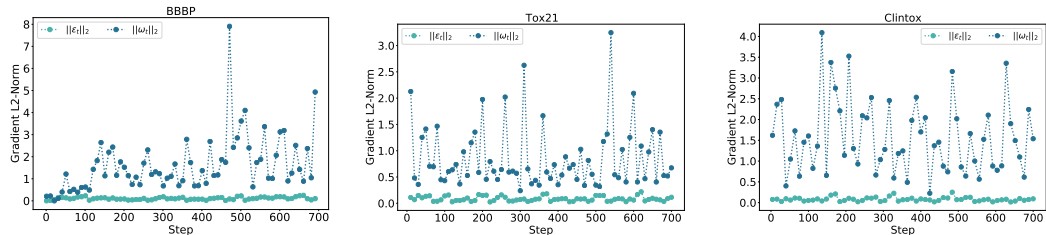

Figure 9: Illustration on the observation of $||\epsilon_t||_2$ and $||\omega_t||_2$ during training on GROVER with three datasets.

Regarding the attention matrix over the CoMPT model on BBBP dataset, we observe that models optimized with Adam, GraphSAM, and SAM obtain average entropies of 2.4130, 2.7239, and 2.8382, respectively. That means the application of SAM approaches can smooth the attention scores to avoid overfitting. The attention heat map is shown in Fig. 10.



Figure 10: Attention heat map.

In CV, the model's perturbation gradient variations are often consistent with updating gradients, which is not in line with the phenomena on the molecular graphs. To further address the concern and provide holistic discussion, we add the following experiments to compare the performance of lookSAM, AE-SAM, and GraphSAM in the CV and molecule datasets, respectively. LookSAM and GraphSAM are efficient SAMs designed based on the gradient varification patterns of models in their respective domains, and AE-SAM is an adaptively-updating efficient method according to the comparison evaluation between gradient norm and a pre-defined threshold.

Table 11: Performance of various SAM methods in the CV domain.

| ResNet-18 (CV) | CIFAR-10 | CIFAR-100 |
|---|---|---|
| +SGD | 95.41 | 78.21 |
| +SAM | **96.53** | 80.18 |
| +LookSAM | 96.28 | 79.91 |
| +AE-SAM | 96.49 | **80.31** |
| +GraphSAM | 95.86 | 78.69 |

As shown in Fig.11 and Fig. 7, it is observed that LookSAM has better results on the image datasets instead on the molecular graphs investigated in this work. On the contrary, our GraphSAM works on the molecules but leads to the worst performances on the image benchmarks. That is because model gradient variations are diverse across the different domains. It is challenging to transfer the efficient SAMs designed based on the specific gradient patterns. AE-SAM achieves an acceptable performance on the molecular graphs since the perturbation gradient norms of graph transformers are monitored to inform the necessary gradient re-computation. But it is not as good as GraphSAM where we accurately fit the perturbation gradients at each step. In summary, GraphSAM is optimized particularly for graph transformers based on the empirical observations of gradient variations.

