# OpenReview forum: "Efficient Sharpness-Aware Minimization for Molecular Graph Transformer Models"
_ICLR.cc/2024/Conference — ICLR 2024 poster_

### Official Review · Reviewer_1va4 · 2023-10-22

**Soundness:** 3 good
**Presentation:** 2 fair
**Contribution:** 3 good
**Rating:** 6
**Confidence:** 4

**Summary:**

This paper designs a computation efficient variant of sharpness aware minimization (SAM) for molecular graph transformers.

**Strengths:**

S1. This paper observes that flat minima generalize better for molecular graph transformers. This extends the boundary of sharpness awareness to a new domain.

S2. To overcome the computational burden, this paper leverages domain specific knowledge to design GraphSAM. Supporting theories are derived to explain the reasons behind design.

S3. GraphSAM improves throughput with comparable performance with SAM. Compared to other variants of computation efficient SAM, GraphSAM performs better in the considered setting.

**Weaknesses:**

W1. Can the authors elaborate more on equation (3)? In particular, are there any specific reasons for choosing $\epsilon_t$ as the moving average of $w_t/\|w_t \|$? What will happen if $\epsilon_t$ becomes moving average of $w_t$?

W2. What are the specific reasons behind the choice of StepLR for the $\rho$-schedulers? Does other approaches, such as cosine schedulers or inverse square root schedulers help?

**Questions:**

Q1. Table 1 does not fit into the paper properly.

Q2. Some of the expressions are unnecessarily complicated. For example, in equation (2) and (3), $sign(\epsilon_t)|\epsilon_t|$ can be replaced by $\epsilon_t$.

Q3. Missing references on SAM variants.

[1] Du, Jiawei, Daquan Zhou, Jiashi Feng, Vincent YF Tan, and Joey Tianyi Zhou. "Sharpness-Aware Training for Free." arXiv preprint arXiv:2205.14083 (2022).

[2] Li, Bingcong, and Georgios B. Giannakis. "Enhancing Sharpness-Aware Optimization Through Variance Suppression." arXiv preprint arXiv:2309.15639 (2023).

---

> ### Author Response · Authors · 2023-11-21
>
> We appreciate a lot for your careful review, and would like to provide responses to your mentioned questions and weaknesses one by one.
>
> ###**[W1 - What will happen if $ε_{t+1}$ becomes moving average of $w_{t}$? — It can cause gradient explosion.]**
>
> Thank you for your insightful observation. Utilizing $|w_{t}|$ without normalization could make $ε_{t+1}$ grow excessively large, which might trigger a gradient explosion. Such an occurrence would critically disrupt the training process, rendering it not only ineffective but also at risk of divergence.
> In Eq. (3), the choice to use $w_{t}/|w_{t}|$ for updating $ε_{t+1}$ is deliberate and crucial for the stability and effectiveness of the update process. This formulation ensures that the update direction of $ε_{t+1}$ is determined by $w_{t}$ while its magnitude is controlled by the factor (1-β). By normalizing  $w_{t}$ with its magnitude $|w_{t}|$ , we prevent the scale of $w_{t}$ from disproportionately influencing the update of $ε_{t+1}$. If we were to use $w_{t}$ directly for updating $ε_{t+1}$, the absence of normalization would lead to significant issues. Our experiments have demonstrated that $|w_{t}|$ >> $|ε_{t}|$, further underscoring the necessity of this normalization step.
>
> ###**[W2 - Does Cosine schedulers or inverse square root schedulers help? — They have some effect, but StepLR is the best.]**
>
>
> Thank you for your question regarding the choice of the StepLR scheduler for the ρ-schedulers in our model. The decision to use StepLR was primarily guided by its simplicity and effectiveness in controlling the learning rate in discrete steps. This allows for a straightforward reduction of the ρ size at predetermined epochs, which we found to be beneficial for the steady convergence to a flat region of our model's parameters. In response to your suggestion, we have added the following two sets of experiments:
>
> CosineLR: $ρ_{new}$=$\frac{ρ_{initial}}{2}*(1+cos(\frac{epoch·π}{EPOCH}))$
>
> InverseSquareLR: $ρ_{new}$=$\frac{ρ_{initial}}{\sqrt{max(epoch,k)}}$
>
> where epoch is the current training epoch, EPOCH is total training epoch, and k=1, $ρ_{initial}$ = 0.05.
>
> |GROVER+GraphSAM | BBBP(ROC-AUC↑) | Tox21 (ROC-AUC↑)| Sider (ROC-AUC↑)| Clintox(ROC-AUC↑)|ESOL(RMSE↓) | Lip(RMSE↓)|
> |   :----: |   :---: |    :---: |  :---: |    :---: |    :---: |    :---: |
> | Original model| 0.917 | 0.822| 0.649 |0.853|0.639 | 0.671|
> | +StepLR| **0.928** | **0.846**| **0.665** |**0.866**|**0.625** | **0.654**|
> | +CosineLR | 0.921 | 0.828| 0.651|0.856| 0.633|0.661
> | +InverseSquareLR | 0.924 | 0.836 |0.654|0.858|0.628|0.665
>
> We acknowledge that other approaches, such as cosine schedulers or inverse square root schedulers, could also be effective.  While these alternative schedulers present intriguing possibilities, our initial experiments with StepLR showed the most promising results, which led to its selection for the current study. However, we recognize the value in exploring these alternative scheduling methods in future work, as they may offer different advantages in specific training contexts or with certain types of datasets.
>
>
> ###**[Q1 - Table 1 does not fit into the paper properly. — Fixed, and thank you for the close read!]**
>
>
> ###**[Q2 - $sign(ε_{t})|ε_{t}|$ can be replaced by $ε_{t}$! — Yes, you are right and we fixed it.]**
>
> ###**[Q3 - Missing references on SAM variants. — We have added them.]**
>
> Thank you for highlighting the omission of references related to SAM variants in our manuscript. In our revised manuscript, we will include and discuss the two SAM variants [1, 2] in the related work.
>
> **References:**
>
> [1] Du, Jiawei, Daquan Zhou, Jiashi Feng, Vincent YF Tan, and Joey Tianyi Zhou. "Sharpness-Aware Training for Free." arXiv preprint arXiv:2205.14083 (2022).
>
> [2] Li, Bingcong, and Georgios B. Giannakis. "Enhancing Sharpness-Aware Optimization Through Variance Suppression." arXiv preprint arXiv:2309.15639 (2023).

---

### Official Review · Reviewer_mSQh · 2023-10-26

**Soundness:** 3 good
**Presentation:** 2 fair
**Contribution:** 3 good
**Rating:** 6
**Confidence:** 5

**Summary:**

The paper proposes GraphSAM to reduce the training cost of sharpness-aware minimization (SAM) and improve the generalization performance of graph transformer models. GraphSAM uses the updating gradient of the previous step to approximate the perturbation gradient at the intermediate steps smoothly and theoretically proves that the loss landscape of GraphSAM is limited to a small range centered on the expected loss of SAM. Extensive experiments on six datasets with different tasks demonstrate the effectiveness and efficiency of GraphSAM.

**Strengths:**

- **Motivation is clear**: SAM is a powerful optimizer but doubles the computational cost compared with the base optimizer. Thus, **improving its efficiency** (the focus of this paper) is an important problem
- GraphSAM improves the efficiency of SAM by **approximating the perturbation gradient** $\epsilon_{t+1}$ as the moving average of $\epsilon_t$ and $\omega_{t}$ based on two observations: (i) $\epsilon_t$ is close to $\epsilon_{t+1}$ (Figure 3(a)) and (ii) $\epsilon_{t+1}$ is close to $\omega_t$ (Figure 3(c))
- By approximating the perturbation gradient, GraphSAM almost **does not need to compute the perturbation gradient** (only once per epoch). Thus, GraphSAM is efficient
- Experimental results on the graph dataset show that GraphSAM is comparable with SAM (Tables 1 and 2) and is more efficient (Table 2)

**Weaknesses:**

- **writing**:
  - The colors in Figure 1 are inconsistent with the observations: "compared to the base optimizer of Adam (the blue bar)", but the blue is LookSAM? also, SAM (cyan)?
  - Figure 1, it is better to include the proposed GraphSAM as well
  - Related works: "most of the work still ignores the fact of SAM’s double overhead (Damian et al., 2021; Kwon et al., 2021;Wang et al., 2022)." I think some methods have attempted to mitigate this issue, e.g., AE-SAM, SS-SAM, better to discuss them here
  - Section 3, "SAM consumes double overhead due to the extra computation of perturbation gradient compared with the base optimizer." existing methods (AE-SAM, SS-SAM) have tried to mitigate this issue, better to discuss them here, and why they cannot be used for graph datasets, this is also the motivation for the proposed GraphSAM
  - Figure 3, sub-caption for each subfigure
  - Table 1: also compare with GROVER+AE-SAM/SS-SAM
  - "AE-SAM and RST periodically compute the update gradient by different strategies": AE-SAM is an adaptive strategy, not a periodic strategy
- it seems that the proposed GraphSAM is general, not limited to graph transformers, and can also be used in computer vision. Thus, it is better to conduct some experiments on computer vision.
- ablation study for hyperparameters $\gamma, \lambda, \beta$
- Theorem 2, based on the proof, the proportion ratio depends on the $\theta$ (Eq(7)), it is trivial. if we assume the loss is Lipschitz, then the constant can be independent of $\theta$.

**Questions:**

see the questions in weakness.

======

I have read the rebuttal and other reviews. Most of my concerns are resolved, so I maintain my score of 6.

---

> ### Author Response · Authors · 2023-11-21
> **Response 1/2**
>
> Thank you for the many valuable comments on writing, we have also been corrected! Notably, the SS-SAM you mention is the same paper as the RST compared in our paper.
>
> ###**[W1.1 - The colors in Fig.1 are inconsistent — Fixed, and thank you for the close read! ]**
>
> ###**[W1.2 - Add GraphSAM in Fig.1 — We have added GraphSAM in Fig.1.]**
>
> ###**[W1.3 and W1.4 - Discuss existing efficient methods, e.g., AE-SAM and SS-SAM. — We have discussted and compared them in Appendix.]**
>
> Indeed, we have provided a detailed comparison of these efficient SAM versions, including AE-SAM and SS-SAM(RST), in Appendix A.4 and A.5. We specifically discuss why these methods are less suited for molecular graph datasets and how GraphSAM is uniquely designed to address the challenges inherent to such data.
>
> For AE-SAM, the trend of its squared stochastic gradient norms in CV domain is not consistent with the molecule data. This leads to a decrease in its sensitivity, which is not as good as its performance in CV, but works better than other methods.
>
> For SS-SAM, the advantage is efficiency; but the disadvantage is obvious: random decisions about whether or not to update the gradient can lead to performance reduction. Its performance in CV literature is also poor.
>
> ###**[W1.5 - sub-caption for each subfigure. — We put the sub-caption for each subfigure uniformly into the description.]**
>
> ###**[W1.6 - Table 1: also compare with GROVER+AE-SAM/SS-SAM. — We have compared with CoMPT and Grover (+ every efficient SAM) in detail in Table 2 and Table 7.]**
>
> ###**[W1.7 - Description of AE-SAM is incorrect. — Fixed, and thank you for the close read!]**
>
>
> ###**[W2 - GraphSAM can also be used in CV? — Yes, but it does not perform ideally.]**
>
> Sorry, GraphSAM doesn't perform well in the CV domain. It is experimentally found that GraphSAM is a good approximation for graph transformers rather than in CV domain. Although our proposed method is designed based on the empirical observations, actually, Observations 1 and 2 are presented only in the graph transformers, while they do not appear in those of CV domain [1]. In CV, the model's perturbation gradient variations are often consistent with updating gradients, which is not in line with the phenomenons on the molecular graphs.
>
> To further address the concern and provide holistic discussion, we add the following experiments to compare the performance of lookSAM [1], AE-SAM and GraphSAM in the CV and molecule datasets, respectively. LookSAM and GraphSAM are efficient SAMs designed based on the gradient varification patterns of models in their respective domains, and AE-SAM is an adaptively-updating efficient method accoding to the comparison evaluation between gradient norm and a pre-defined threshold.
>
> |ResNet-18 (CV) | CIFAR-10 | CIFAR-100|
> |   :----: |   :---: |    :---: |
> | +SGD| 95.41 | 78.21|
> | +SAM| **96.53** | 80.18|
> | +LookSAM | 96.28 | 79.91|
> | +AE-SAM |96.49 | **80.31**|
> | +GraphSAM | 95.86 | 78.69 |
>
>
> |CoMPT (Graph) | Tox21 | Sider| ClinTox|
> |   :----: |   :---: |    :---: |   :---: |
> | +Adam| 82.81 | 62.17| 91.41|
> | +SAM| 83.96 | 64.33| 92.73|
> | +LookSAM | 82.55| 62.54| 91.64|
> | +AE-SAM | 83.33| 63.16| 91.92|
> | +GraphSAM | **84.11** | **64.58** |**93.78** |
>
>
> It is observed LookSAM has better results on the image datasets instead on the molecular graphs investigated in this work. On the contrary, our GraphSAM works on the molecules but leads to the worst performances on the image benchmarks. That is because model gradient variations are diverse accross the different domains. It is challenging to transfer the efficient SAMs designed based on the specific gradient patterns. AE-SAM achieves an acceptable performance on the molecular graphs since the perturbation gradient norms of graph transformers are monitored to inform the necessarity of gradient re-computation. But it is not as good as GraphSAM where we accurately fit the perturbation gradients at each step. In summary, GraphSAM is optimized particularly for the graph transformers based on the empirical observations of gradient variations.
>
> **References:**
>
> [1] Yong Liu, Siqi Mai, Xiangning Chen, Cho-Jui Hsieh, Yang You. Towards efficient and scalable sharpness-aware minimization. CVPR 2022.

---

> > ### Author Response · Authors · 2023-11-21
> > **Response 2/2**
> >
> > ###**[W3 - Ablation study for hyperparameters β, γ,λ ? — We have included experiments for hyperparameter analysis in the Appendix.]**
> >
> > Thank you for your inquiry regarding the ablation study for hyperparameters. Recognizing the length constraints of the main paper, we have included detailed experiments on hyperparameter ablation in Appendix with Table 3 for β, Table 9 for γ and Fig.7 for λ. These sections in the appendix offer a comprehensive view of how various hyperparameters influence the performance and behavior of our proposed method.
> >
> >
> > ###**[W4 - If we assume the loss is Lipschitz, then the constant can be independent of $θ$? — Your assumption is correct, but does not solve the problem in SAM.]**
> >
> > The essential nature of using a first-order Taylor expansion involves calculating the gradient of the current parameter θ, which aims at more than just θ. Lipschitz continuity of a function provides a bound on how much the function can change between two points. While it's a valuable property, especially in ensuring stability and robustness of optimization algorithms, it doesn't directly provide a method to find flat minima. In the context of SAM and GraphSAM, the Lipschitz constant would tell us the maximum rate at which the loss can change, but it doesn't directly facilitate the optimization towards flat regions of the loss landscape.
> >
> > Both Sharpness-Aware Minimization (SAM) and its variants use a first-order Taylor expansion in their formulations, primarily because it offers a practical and computationally efficient way to approximate the worst-case loss in a neighborhood around the current parameter values.
> >
> > Consequently, we still choose the first-order Taylor expansion as a derivation condition in this paper, instead of assuming the loss is Lipschitz.

---

### Official Review · Reviewer_sT1K · 2023-10-26

**Soundness:** 3 good
**Presentation:** 3 good
**Contribution:** 3 good
**Rating:** 6
**Confidence:** 3

**Summary:**

The study introduces Sharpness-aware minimization (SAM) in computer vision, a technique effective in eliminating sharp local minima in training trajectories and countering generalization degradation. However, its dual gradient computations during optimization increases time costs. To address this, a novel algorithm named GraphSAM is proposed. It lowers the training expenses of SAM while enhancing the generalization performance of graph transformer models. The approach is underpinned by two main strategies: gradient approximation and loss landscape approximation. Empirical tests across six datasets validate GraphSAM's superiority, particularly in refining the model update process. Anonymized code is also provided for further reference.

**Strengths:**

(1) The paper is exceptionally coherent in its writing. The content is presented seamlessly, and the expression is clear.

(2) This paper is both conceptually and technically innovative. It uniquely reutilizes the updating gradient from the previous step, approximating the perturbation gradient in an efficient manner.

(3) The experimental design of this study is well-conceived. It systematically validates the representational capabilities of GraphSAM. Additionally, the appendix offers an extensive set of supplementary experiments.

**Weaknesses:**

(1) In Observation 1, the author notes the differing changes of the perturbation gradient and the updating gradient throughout the training process. I wonder, is this a general phenomenon? Could more case studies be provided to illustrate this further?

(2) The results presented in Table 1 seem to show limited improvement on certain datasets. I suggest the author consider incorporating an efficiency aspect into the table. Efficiency is a crucial contribution, yet it hasn't been emphasized adequately in the experimental section, which seems inconsistent.

**Questions:**

see weakness

---

> ### Author Response · Authors · 2023-11-21
>
> We appreciate a lot for your careful review, and would like to provide responses to your mentioned questions and weaknesses one by one.
>
>
> ###[**W1 -  Is the differing changes of gradient a general phenomenon? — Yes, it is.]**
>
> The observation of differing changes in the perturbation gradient and the updating gradient throughout the training process is indeed a general phenomenon. To further substantiate this, we have enriched the Figure 8 in Appendix 8 with additional case studies illustrating these variations. They are consistent with the conclusions in Observation 1.
>
>
> ###[**W2 -  Efficiency is a crucial contribution, add experiments to illustrate — Agreed, we have compared the efficiency in Table 2 and Table 7.]**
>
> We have included comprehensive efficiency comparisons in Table 2 of the main paper and Table 7 of Appendix A.5. These tables specifically highlight the efficiency aspects of GraphSAM, offering a balanced view of both its performance and computational efficiency.

---

### Official Review · Reviewer_QvPb · 2023-11-01

**Soundness:** 3 good
**Presentation:** 4 excellent
**Contribution:** 3 good
**Rating:** 6
**Confidence:** 4

**Summary:**

The authors study Sharpness Aware Minimization (SAM), an effective training objective in the image domain to avoid "sharp minima", in the context of graph transformers for molecular data. The proposed efficient SAM-alternative, termed GraphSAM, empirically performs on par with SAM while being more efficient.

**Strengths:**

1. More efficient approach for SAM (roughly 30%-50% faster training)
1. GraphSAM consistently improves the performance of the base model
1. Approach is ablated and rich explanations are provided
1. The paper is well organized and it is easy to follow.

**Weaknesses:**

1. The relation to graph machine learning is a bit obscure: is GraphSAM only a good approximation for graph transformers? After all, GraphSAM does not rely on anything graph-related besides the observations made with SAN + graph transformers. Although it is fine to focus on molecular data, the paper would benefit significantly from a holistic discussion (and some rudimentary experiments).
1. The theoretical statements could be better motivated and more decisively embedded into the story. It remains a bit vague what the actual implications are. For example, the conclusion of Theorem 1 is speculative "by replacing the inner maximum of LG(θ + εˆS) with the maximum of LG(θ + εˆG), we tend to smooth the worse neighborhood loss in the loss landscape. In other words, if one could minimize the upper bound given by LG(θ + εˆG), the final loss landscape is as smooth as that obtained from SAM." Alternatively, the wording may be improved. It is not clear why optimizing the upper bound necessarily results in a "loss landscape is as smooth as that obtained from SAM".
1. Proof of Theorem 1 relies on empirical observations and (from my perspective) strong assumptions. However, the authors do sufficiently not discuss this in the main part and, thus, the current presentation is misleading. In other words, these assumptions, etc. should be made explicit in a rather prominent way in the main part (e.g. "and after ||ω0 ||2 experimental analysis, the updating gradient ω ≫ ε " is not stated clearly in main part). I think it would be better to drop the term "Theorem" and rather give some intuitive, mathematically motivated explanation.

Minor:
1. Missing space in the first line of page 2
1. The bold highlighting in Table 1 is counter-intuitive. Perhaps add an additional marker to highlight the best model per task.

**Questions:**

1. Is GraphSAM only a good approximation for graph transformers? How is GraphSAM working, e.g., in the image domain?
1. Are there any observations of how the behavior of the graph transformer changes if trained with GraphSAM? For example, do the attention scores then align better with the graph connectivity, or do they become smoother (higher entropy)?
1. Is it possible to compare GraphSAM with other efficient SAM derivates besides SAM-k (e.g., see related work section)?
1. Is The proportionality in Theorem 2 not merely caused by the first-order Taylor approximation (Eq 6 in A1.1)? How do the authors know that the linear term is a sufficient approximation?

---

> ### Author Response · Authors · 2023-11-21
> **Response 1/2**
>
> We appreciate a lot for your careful review, and would like to provide responses to your mentioned questions and weaknesses one by one.
>
> ###**[W1 & Q1 - Is GraphSAM only a good approximation for graph transformers? How is GraphSAM working, e.g., in the image domain? — Yes, GraphSAM only works good at molecular graph domain.]**
>
>
> It is experimentally found that GraphSAM is a good approximation for graph transformers rather than in CV domain. Although our proposed method is designed based on the empirical observations, actually, Observations 1 and 2 are presented only in the graph transformers, while they do not appear in those of CV domain [1]. In CV, the model's perturbation gradient variations are often consistent with updating gradients, which is not in line with the phenomenons on the molecular graphs.
>
> To further address the concern and provide holistic discussion, we add the following experiments to compare the performance of lookSAM [1], AE-SAM and GraphSAM in the CV and molecule datasets, respectively. LookSAM and GraphSAM are efficient SAMs designed based on the gradient varification patterns of models in their respective domains, and AE-SAM is an adaptively-updating efficient method accoding to the comparison evaluation between gradient norm and a pre-defined threshold.
>
> |ResNet-18 (CV) | CIFAR-10 | CIFAR-100|
> |   :----: |   :---: |    :---: |
> | +SGD| 95.41 | 78.21|
> | +SAM| **96.53** | 80.18|
> | +LookSAM | 96.28 | 79.91|
> | +AE-SAM | 96.49 | **80.31**|
> | +GraphSAM | 95.86 | 78.69 |
>
>
> |CoMPT (Graph) | Tox21 | Sider| ClinTox|
> |   :----: |   :---: |    :---: |   :---: |
> | +Adam| 82.81 | 62.17| 91.41|
> | +SAM| 83.96 | 64.33| 92.73|
> | +LookSAM | 82.55| 62.54| 91.64|
> | +AE-SAM | 83.33| 63.16| 91.92|
> | +GraphSAM | **84.11** | **64.58** |**93.78** |
>
>
> It is observed LookSAM has better results on the image datasets instead on the molecular graphs investigated in this work. On the contrary, our GraphSAM works on the molecules but leads to the worst performances on the image benchmarks. That is because model gradient variations are diverse accross the different domains. It is challenging to transfer the efficient SAMs designed based on the specific gradient patterns. AE-SAM achieves an acceptable performance on the molecular graphs since the perturbation gradient norms of graph transformers are monitored to inform the necessarity of gradient re-computation. But it is not as good as GraphSAM where we accurately fit the perturbation gradients at each step. In summary, GraphSAM is optimized particularly for the graph transformers based on the empirical observations of gradient variations.
>
>
> ###**[W2 - The conclustion of Theorem 1 is not clear. — Will add to the following new explanations.]**
>
> Thank you for your constructive feedback regarding the theoretical aspects of our paper. We hope the following revisement can mitigate your concerns.
> The revised sentence is: "Recalling the min-max optimization problem of SAM in Eq. (1), if we replace the inner maximum objective from $L_{G}(θ + ε^{S})$ to $L_{G}(θ + ε^{G})$, the graph transformer is motivated to smooth a worse neighborhood loss in the loss landscape. In other words, the proposed GraphSAM aims to minimize a rougher neighborhood loss, whose value is theoretically larger than that of SAM, and obtain a smoother landscape associated with the desired generalization."
>
> ###**[W3 - Proof of Theorem 1 relies on empirical observations and strong assumptions. — Agreed, we have now changed it.]**
>
> Thanks for your suggestions, and we have uniformly changed 'Theorem' to 'Conjecture'. Conjecture is a statement or conclusion based on insufficient evidence, rather than through rigorous proof. We rewrite Theorem 1 to include the experimental assumption that supports our conclusion.
>
> The new  Conjecture 1 is:
> Let $ϵ̂_{S}$ and $ϵ̂_{G}$
> denote the perturbation weights of SAM and GraphSAM, respectively,
> where we ignore the subscript of t for the simple representation.
> Suppose that  ω/|ω| >>  ϵ ,as  empirically  discussed in Observation 1, and
> $|ϵ̂_{S}|$ < $|ϵ̂_{G}|$
> for ρ> 0,
> designating $ϵ̂_{S}$ as the ground-truth. We have:
>
> L_{G}(θ+$ϵ̂_{S}$)
>  ≤
>  L_{G}(θ+$ϵ̂_{G}$)
>
> **References:**
>
> [1] Yong Liu, Siqi Mai, Xiangning Chen, Cho-Jui Hsieh, Yang You. Towards efficient and scalable sharpness-aware minimization. CVPR 2022.

---

> ### Author Response · Authors · 2023-11-21
> **Response 2/2**
>
> ###**[M1 - Missing space — Fixed, and thank you for the close read! ]**
>
> ###[**M2 - Counter-intuitive bold highlighting — Sure, we have changed it]**
>
> In the revised manuscript, we maintain the bold highlighting for the best optimizer of graph transformer per task, but also add 'underline' as an additional marker to further emphasize the top-performing model in each category.
>
>
> ###[**Q2 - How the behavior of the graph transformer changes if trained with GraphSAM? — When the model trained with GraphSAM, nodes' attention score matrix average entropy of same molecular graph is higher.]**
>
> This is a wonderful suggestion. Regarding the attention matrix over BBBP dataset, we observe that models optimized with Adam, GraphSAM, and SAM obtain average entropies of 2.4130, 2.7239, and 2.8382, respectively. That means the application of SAM approaches can smooth the attention scores to avoid the overfitting. We have added the attention heat map analysis in appendix. In addition, Figure 5 at main paper shows that incorporating GraphSAM leads to smoother training and testing loss curves and reduces the gap between them. This smoother convergence suggests that GraphSAM helps learning a more robust and stable representation, which indicates a more evenly distributed attention across the graph's features.
>
>
>
> ###[**Q3 -  Compare GraphSAM with other efficient SAM derivates? — Yes, we have compared with them, and GraphSAM is both efficient and high performing.]**
>
> We have compared with other efficient SAMs (e.g., LookSAM, AE-SAM, and RST) in Table 2 of the main paper and Table 7 of Appendix A.5. We observe GraphSAM is both efficient and high-performaning on molecular graphs, while the other efficient SAMs deliver poor generalization results. The detials can be checked in paper.
>
> ###[**Q4 -  Is The proportionality in Theorem 2 not merely caused by the first-order Taylor approximation? How do the authors know that the linear term is a sufficient approximation? — Yes, we leverage the first-order Taylor approximation, but it is accurate enouhgh.]**
>
> The proportionality in Theorem 2 is indeed analyzed mainly according to the first-order Taylor approximation. We are aware that although the higher-order terms could offer a more nuanced view, the first-order term often dominates in practical scenarios, making it a sufficient approximation for the purpose of our analysis. Take the second-order Taylor expansion as an example:
>
> $L_{G}$(θ+ϵ̂) ≈$L_G(θ)$ + ϵ̂$\nabla_\theta$ $L_G(θ)$ +$ \frac{1}{2} ϵ̂^{2}H(θ)$,
>
> where $\mathbf{H}(\theta)$ is a Hessian matrix (i.e., a matrix of second-order derivatives) of the function with respect to $\theta$. $\hat{\epsilon}$ is a mapped perturbation gradient obtained by Eq. (2) and $||\hat{\epsilon}||_{2}$ is pretty small, which is often at the scale $10^{-3}$. Therefore the second-order expansion term  as well as the laters can be omitted. Taylor's first-order expansion can be sufficient approximation.

---

> > ### Comment · Reviewer_QvPb · 2023-11-23
> > **Follow-up**
> >
> > I thank the authors for the careful and elaborate response.
> >
> > My points have been addressed. The authors, though, could elaborate more in the main part why GraphSAM is only a good approximation for molecular tasks.

---

> ### Author Response · Authors · 2023-11-23
>
> Dear Reviewer QvPb,
>
> We sincerely appreciate your thoughtful feedback. In response to your suggestion, we elaborate in Observation 1 of the main text on why GraphSAM is only a good approximation specifically for molecular tasks. Meanwhile, we compare GraphSAM with other efficient SAMs designed for the CV domain and analyze their limitations in the molecular graph domain in detail as shown in Appendix A.4.
>
> Best regards,
>
> The Authors of ’Efficient Sharpness-Aware Minimization for Molecular Graph Transformer Models‘

---

### Author Response · Authors · 2023-11-21
**Revision Summary**

We thank all the reviewers for their time and effort in helping us improve the quality of the paper. We are glad that all reviewers agreed our method GraphSAM on GraphTransformer is efficient and effective. All reviewers also recognized the contribution of GraphSAM in the domain of molecular graphs.

We have updated the paper to incorporate constructive suggestions. We summarize the major changes, and make the changed text blue in the main manuscript:

1. [QvPb] Updated the description and explanation of Theorem 1. Also changed the word 'Theorem' to 'Conjecture' in **the main manuscript**.
2. [QvPb, 1va4, sT1K] Additional experimental observations of perturbation gradient $\epsilon$ and updating gradient $\omega$ in **Appendix A8**.
3. [QvPb] Additional attention heat map experiments with Adam, SAM, and GraphSAM in **Appendix A8**.
4. [QvPb, mSQh] Additional experiments of GraphSAM in CV domain in **Appendix A8**.
5. [QvPb, 1va4, mSQh] Modifications to Figure 1 and Table 1 and some writing issues of **the main manuscript**.

---

### Meta-Review · Area_Chair_YTTX · 2023-12-09

**Metareview:**

By incorporating lessons from an empirical analysis of optimization dynamics in Graph Transformers and SAM in molecular structure prediction tasks, this paper develops a new variant of SAM suitable for this domain. The new variant (GraphSAM) utilizes the observed gradient dynamics to improve the computational efficiency of SAM. The proposed algorithms is supported by some theoretical arguments (although under strong assumptions) but the empirical verification of the method is reasonably covered (multiple datasets+ablation study). The authors have also shared the code of their algorithm. The paper is an interesting example of leveraging domain specific knowledge to design a new variant of SAM.

The reviewers find the approach interesting and the empirical performance impressive. There were some clarification questions during rebuttal, for which authors replied with additional experiments (including results of GraphSAM and other methods on vision tasks) and revised parts of the paper to address these concerns. All reviewers are leaning toward accept. Reviewer 1va4 responded that "The concerns are discussed in detail, and addressed". However, I did not see a change in their score, and they were non-responsive despite reminders, thus unclear if they were willing to increase their initial score.

The paper is borderline accept, but given that reviewers's recommendation are all on the accept side, and they found most of their concerns addressed, and also because the paper presents strong empirical performance I recommend accept. Please incorporate the feedback from reviewers and clarifications you provided in the rebuttal into the final version of the paper.

**Justification For Why Not Higher Score:**

The theoretical arguments could have been more rigorous.

**Justification For Why Not Lower Score:**

Reviewer's recommendation is consistently on the accept side, and the empirical results of the paper are impressive.

---

### Decision · Program_Chairs · 2024-01-16

Accept (poster)